

# Spatial and temporal changes in autumn Eurasian snow cover and its relationship with the Arctic Oscillation

Gareth J. Marshall[1]

[1]British Antarctic Survey, Cambridge, CB3 0ET, UK

*Correspondence to*: Gareth Marshall (gjma@bas.ac.uk)

**Abstract.** Previous studies have demonstrated that variations in the seasonal expansion of Eurasian snow cover (SC) can influence the following winter Arctic Oscillation (AO) and, consequently, affect mid-latitude weather. We examine changes in the extent and rate of autumn Eurasian SC advance and the temporal variability of the magnitude and sign of the SC-AO relationship. Novel aspects are (i) the use of the latest version of the 20[th] Century Reanalysis (20CR), allowing analysis back

to 1836; (ii) adjusting the reanalysis SC through comparison with observations; and (iii) investigating spatial variation in the frequency of significant SC-AO relationships across Eurasia.

Over the past 50 years the snow advance indices (SAI) demonstrate a slowing and accelerating of snow advance in October and November ($p < 0.01$), respectively, corresponding to a greater contemporaneous decrease in SC extent in October than November and thus a postponement of SC onset. The most temporally robust spatial SC-AO relationship is a longitudinal

dipole such that positive (negative) relationships between October SAI and the AO are more frequent in western (eastern) Eurasia. As the sum of the two regional correlations closely matches the correlation for Eurasia as a whole, an especially strong October SAI-AO relationship occurs when the sign of the relationship in one of these regions is reversed from climatology. Future work will aim to determine the exact linkages behind this new finding in the context of contemporaneous changes in regional atmospheric circulation and snow cover and the many additional factors observed to influence the SC-AO relationship.

## 1 Introduction

It has long been known that the Arctic has been warming significantly faster than the global average, but a recent study has revealed that this 'Arctic amplification' is even greater than previously thought, being a factor of four over the past half-century (Rantanen et al., 2022). Seasonally, Arctic amplification is strongest during October-December, in response to enhanced

upwelling of heat and momentum into the atmosphere from newly opened areas of open water. This seasonal maximum coincides with the onset of the remarkable autumnal expansion of Eurasian snow cover (SC) (Fig. 1), an important and rapidly changing component of the Earth's climate system. SC has a local climate impact, resulting from its high surface reflectivity, thermal emissivity and ground-insulating properties (e.g., Henderson et al., 2018). It can have a significant effect on both the



socio-economic activities of Arctic societies, often via natural hazards such as avalanches and flooding, and the natural
environment (e.g., Callaghan et al., 2011).

Snowfall depends on the regional concurrence of sufficient concentrations of water vapour within the atmosphere together
with suitably low temperature and pressure, while at ground-level, SC is contingent on low surface temperatures and an absence
of scouring winds. (e.g., Mudryk et al. 2017; Allchin and Déry, 2020). There is often an east-west dipole in SC anomalies
across Eurasia with one pole in eastern Europe and the opposite pole in southeast Siberia and northern Mongolia (e.g.,
Gastineau et al., 2017; Zhang et al., 2023). Clark et al. (1999) stated that SC variability over Europe and south-western Asia
is largely controlled by temperature while in east Asia it is primarily determined by precipitation availability. However, Kitaev
et al. (2002) suggested that in recent decades the role of low air-temperatures in the formation of Eurasian SC has diminished,
while that of high precipitation has increased. Conversely, Mokhov and Parfenova (2021) found an increasingly strong negative
relationship between October SC and SAT across Eurasia as a whole.

It should be noted that the latter analysis utilised the National Oceanic and Atmospheric Administration Climate Data Record
(NOAA CDR) satellite-based dataset (Estilow et al., 2015). Due to the length of this dataset, back to 1966, it has often been
employed in past studies to examine trends in Eurasian SC and demonstrates an increase in autumn SC (e.g., Cohen et al.,
2012). However, comparison against independent datasets has revealed that the upward trend in CDR from 1992 to 2015 is an
artifact resulting from increasing accuracy of the product as higher resolution satellite data became available (Brown and
Derksen, 2013; Mudryk et al., 2017; Urraca and Gobron, 2023). In reality, autumn SC, SC onset (SCO) and snow depth (SD)
have decreased in recent decades across much of Eurasia (e.g., Bulygina et al., 2009; Urraca and Gobron, 2023), consistent
with the observed Arctic warming. Unfortunately, there are often temporal discontinuities in SC datasets assimilating satellite
data and ground-based observations that compromise their ability to provide the temporally homogeneous datasets required
for long-term studies of SC change.

Autumn snow accumulation is associated with regional cyclonic activity. In the western part of the Russian Arctic the weather
systems bringing snowfall often start over the North Atlantic, while in the east they originate from the Bering Sea, Okhotsk
Sea or the North Pacific. Cyclogenesis may also occur locally along the border between different air masses: the very cold air
originating from the climatological continental Siberian High and the Arctic air from the north, which is often warmer and
always more humid than the continental air (Bednorz and Wibig, 2016). In southern Siberia (50-60°N) these cyclones transport
cold Arctic air from the north or even colder Siberian polar air from the east and induce negative temperature anomalies that
allow SC to persist (Bednorz and Wibig, 2017). In addition, Allchin and Déry (2020) suggested that incidences of earlier
Eurasian SC may result from northward advection of moisture by stronger southerly winds, driven by altered gradients in
geopotential height north of the Himalayas. The spatial variability of the different air masses that give rise to the cyclogenesis
of snow-bearing weather systems is primarily associated with low-frequency macro-scale atmospheric circulation variability,



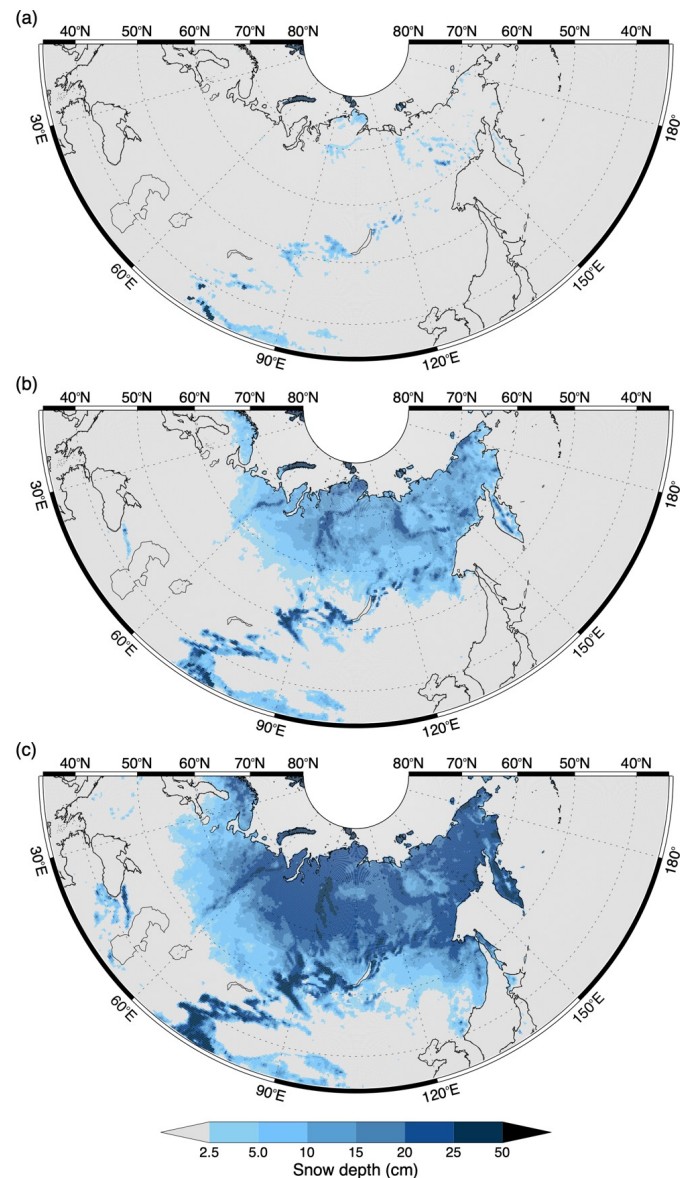

**Figure 1.** Median SD from the European Centre for Medium-Range Weather Forecasts (ECMWF) fifth generation reanalysis (ERA5), 2003-2022 on (a) 30[th] September, (b) 31[st] October, and (c) 30[th] November. Based on the definition used in this analysis, the 2.5 cm SD approximates to SC extent. The SC extent increases from $1.53 \times 10^6$ km$^2$ at the end of September, through $12.77 \times 10^6$ km$^2$ at the end of October to $20.26 \times 10^6$ km$^2$.




which can be expressed as a series of teleconnection patterns that are, in-turn, driven predominantly by either internal atmospheric dynamics or atmosphere-ocean-ice interactions. The two principal teleconnection patterns affecting the autumn Eurasian SC are the Arctic Oscillation (AO) (e.g., Thompson and Wallace, 1998) and the Scandinavian pattern (SCAND) (e.g., Bueh and Nakamura, 2007). We note that, while the North Atlantic Oscillation (NAO) and AO are defined quite differently,

they often have similar climatic impacts across Eurasia (e.g., Yu et al., 2017) and, therefore, in this study we consider them to be the same pattern and simply referred to as the AO.

Arctic amplification and projected climate change impart opposing impacts to snowfall and SC: increased temperatures lead to greater atmospheric humidity and thus more intense snowfall but also to a reduced fraction of solid precipitation and

enhanced snow melting. Future changes in autumn Northern Hemisphere (NH) SC derived from model projections participating in the World Climate Research Programme Coupled Model Intercomparison Project Phase 6 (CMIP6) (Eyring et al., 2015) are broadly linear as a function of the rate of global warming (Mudryk et al., 2020). Based on 20 models, Zhu et al. (2021) concluded that for Eurasia they varied from stabilization at ~2060 at $-10 \pm 5\%$ (relative to 1995-2014) to continuous and ongoing SC losses that reach $-38 \pm 10\%$ by 2100 in the least optimistic climate change scenario. In this last scenario snow

onset across Eurasia is postponed by an average of 23 days (Lin and Chen, 2021). McCrystall et al. (2021) demonstrated that the CMIP6 models show a marked increase in rainfall at the expense of snowfall across the NH in autumn. Nevertheless, Quante et al. (2021) showed that while extreme snowfall events across large areas of the Northern Hemisphere will become less frequent under future warming, they will also become more intense.

In addition to the autumn Eurasian SC having a local climate impact, it may modulate and thus help predict NH climate variability at much larger spatial scales. A substantial scientific literature has focused on the influence that autumn SC variability may have on the following winter AO, an idea first proposed by Cohen and Entekhabi (1999) and which has recently been reviewed in detail by Henderson et al. (2018). The proposed conceptual mechanism through which SC variability is coupled to the atmosphere is summarised in several papers (e.g., Cohen et al., 2007; Cohen et al., 2014b; Furtado et al., 2015;

Henderson et al., 2018). In brief, an expanded October Eurasian SC cools the surface and raises isentropic surfaces in the stable lower atmosphere leading to higher pressure. This pressure anomaly amplifies the regional atmospheric wave pattern such that it favours ridging near the Barents-Kara Seas and Urals and troughing further east, a spatial configuration that provides a local source of upward propagating Rossby waves into the stratosphere. Here, anomalous momentum and heat fluxes are deposited, inducing a westward torque that weakens the polar vortex and can lead to winter stratospheric warming events. The circulation

anomalies subsequently descend into the troposphere over a period of weeks, eventually manifesting as a negative polarity AO. In summary, greater autumn Eurasian SC precedes a negative winter AO that is generally associated with a warmer Arctic but colder and more severe winter conditions in NH mid-latitudes. Case studies of the 2009/10 cold winter demonstrated that



Eurasian SC changes provided significant skill in predicting the record low values of the coincident AO (Cohen et al., 2010; Orsolini et al., 2016).


However, modelling studies that have attempted to reproduce the SC-AO linkages have had mixed success. 'Free-running' models generally fail to replicate the mechanism (e.g., Furtado et al., 2015; Handorf et al., 2015; Peings et al., 2017) while model runs with bias corrections and imposed SC anomalies do better (e.g., Allen and Zender, 2011; Tyrrell et al., 2020). Henderson et al. (2018) concluded that current models are especially poor at capturing the planetary wave generation and stratosphere-troposphere coupling components of the SC-AO mechanism. Further potential causes of the poor model skill include the signal being masked by substantial internal Arctic climate variability (e.g., Peings et al., 2021); the existence of the mechanism having a state dependence such as through the phase of the Quasi-Biennial Oscillation (Garfinkel et al., 2010; Peings et al., 2013; Overland et al., 2016); and other 'competing' but not necessarily independent factors that also impact the AO, such as Arctic Ocean sea ice (Handorf et al., 2015; Delhaye et al., 2024) and Ural Blocking (UB) variability (Peings, 2019).

The Eurasian SC-AO relationship identified by Cohen and Entekhabi (1999) and expanded in successive papers by Cohen and co-authors was necessarily based on a relatively limited period of recent data (e.g.,1973-2012) because of the shortness of the SC record. However, longer time series of spatial SC have subsequently become available as output fields from a number of global reanalyses that extend back to the 19th Century. Using the National Oceanographic and Atmospheric Administration (NOAA) 20th Century Reanalysis (20CR) back to 1900, Peings et al. (2013) demonstrated that the SC-AO relationships observed in the recent period were not temporally invariant (stationary), with the past few decades having the most robust and continuously significant relationships. For most of the 20th Century there was no statistically significant SC-AO relationship. Douville et al. (2016) examined the SC-AO relationship in an additional reanalysis, the European Centre for Medium-Range Weather Forecasts (ECMWF) 20th Century Reanalysis (ERA-20C). While these authors revealed that the temporal variability of the sign and strength of the SC-AO relationship was broadly similar in both reanalyses, Wegmann et al. (2017) established that pre-1950 the two reanalysis products diverted towards different base states with 20CR and ERA20-C having too high or low SD, respectively. In conclusion, Douville et al. (2016) proposed that the temporal modulation of the Eurasian SC-AO relationship might arise simply from stochastic noise and/or that an external physical driver such as the QBO or low-frequency modes of oceanic variability may also be partly responsible.

In this study we (i) examine changes in autumn Eurasian SC, both in terms of its extent and rate of advance (Cohen and Jones, 2011) and (ii) revisit the temporal variability of the magnitude and sign of the SC-AO relationship. There are several novel aspects to our work compared to previous studies. First, the use of the latest version of the 20CR reanalysis (version 3, hereinafter 20CRv3) allows our analyses to go back to 1836. Moreover, in order to reduce bias and detect temporal discontinuities in 20CRv3 SC, we adjust it by comparison with ECMWF fifth generation reanalysis (ERA5) SC data, which





in turn have been validated against SD observations. Moreover, in addition to temporal variability, we investigate whether there is any marked spatial variation in the strength of SC-AO relationships across Eurasia — by subdividing it into subregions (cf. Fig. 2: 5° latitude by 30° longitude). Thus, we can ascertain whether some parts of Eurasia have a consistently strong SC-

AO relationship and the observed non-stationarity results from varying strengths and/or signs in the SC-AO connection across the remainder of the Eurasian domain. Conversely, during periods when there is a statistically significant SC-AO relationship for Eurasia as a whole, is that due to an anomalously strong correlation in those parts of Eurasia where there is generally a relationship of that sign anyway or a more widespread SC-AO relationship of that sign across Eurasia than is usual?

In Sect. 2, we define the five different SC indices examined in this analysis and how they are calculated, describe the different data types employed in this study: observed SD data, reanalysis data and AO indices. In Sect. 3 we explain the methodology used to remove the bias in SC within 20CRv3 and the statistical methods applied in this analysis. The results are presented in two sections: in Sect. 4.1 we examine the trends in the SC indices derived from 20CRv3 while in Sect. 4.2 we explore the spatial and temporal variability in the Eurasian SC-AO relationships. In Sect. 5 we provide some further discussion on one of

the key findings in the study. Finally, in Sect. 6 we summarise our principal conclusions.

## 2 Data

### 2.1 Snow cover indices

In this study we examine trends and Eurasian SC-AO relationships for five different SC indices. Eurasia is defined as the region bounded by 35-80°N and 30-180°E. In order to examine spatial variability in the SC indices and their relationship to

the AO, this region was subdivided into 45 areas, each of 5° latitude by 30° longitude. Note that some of these subregions actually contain no SC data as they are ocean (Fig. 2). The SC indices are listed, together with their acronyms, in Table 1. They comprise three monthly SC extent indices (SCIs) and two monthly SC advance indices (SAIs). The monthly SC extent indices are calculated as the mean of the daily SC data in that month while, following Cohen and Jones (2011), the monthly SC advance indices are computed as the regression coefficient of the least squares linear trend through the daily SC data for

that month. Finally, again following on from Cohen and Jones (2011) and others, the regression coefficient is multiplied by – 1. Therefore, a higher SAI corresponds to a weaker snow advance and vice versa.

In order to calculate the SC, we need to define the SCO. Here, the SCO is defined as the first day of the first period of five or more consecutive days in the snow year (August to July) when the SD is greater than or equal to 2.5 cm. The 2.5 cm definition

for SD was chosen following the analysis of SC fraction and SD by Urraca and Gobron (2023): they found that an SD of 2.5 cm typically equates to an SC fraction of 50%, which is used as the threshold in many analyses of SC extent (e.g., Hori et al. 2017). While the period of five days is shorter than used in some other studies (e.g., Bender et al., 2020), and there will, of course, be some instances of subsequent periods without SC, it matches the definition in other previous analyses of SC (e.g.,



Peng et al., 2013). Moreover, we found that using a longer period than five days meant that there were several years without
an SCO at some of the more southerly meteorological stations. Using the above definition for SCO, the number of times when
there was no SCO was limited to six examples among four stations.

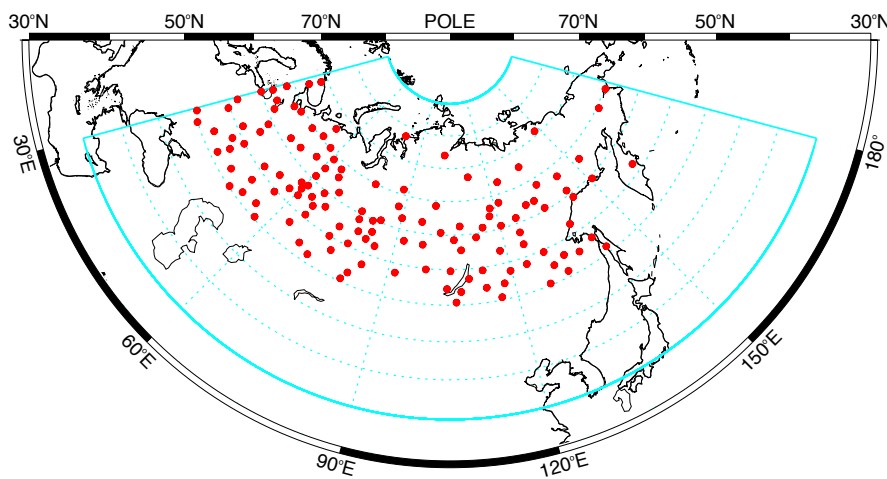

**Figure 2.** Red dots are the location of the 133 stations with daily snow depth observations used to validate ERA5. The thick blue line shows
Eurasia, as defined in this study, while the dashed lines denote the 45 subregions (5° latitude by 30° longitude).

**Table 1.** The snow cover indices examined, and their acronyms used in the paper.

| Snow Cover Index | Acronym |
|---|---|
| Mean snow cover extent in September | SCI_09 |
| Mean snow cover extent in October | SCI_10 |
| Mean snow cover extent in November | SCI_11 |
| Mean rate of snow cover advance in October | SAI_OCT |
| Mean rate of snow cover advance in November | SAI_NOV |


## 2.2 Daily snow depth observations

The daily SD data utilised to test the accuracy of SC in the ERA5 reanalysis were observed at meteorological stations in the
former Soviet Union and comprise part of the Historical Soviet Daily Snow Depth (HSDSD) version 2 dataset (Armstrong,
2001). Today the significant majority of these stations are located within Russia, but some used in this analysis are now situated
in Kazakhstan or Ukraine. The HSDSD dataset spans the time period from 1881 to 1995 so the daily snow depth data were





appended to 2022 using data obtained via the Global Telecommunication System (GTS). These data were acquired from either the UK Met Office Integrated Archive System (MIDAS) Land and Marine Stations Data (Met Office, 2012) or the Spanish Ogimet website. The original measurements were taken as an average from three permanent snow stakes surrounding each station (Ye, 2001), with no changes in procedures since 1965 (Bulygina et al., 2011). The snow depth from the HSDSD stations

is provided to the nearest cm.

While quality control of the HSDSD data has been undertaken (Armstrong, 2001), some 'gross' errors still remain. For example, there are still a small number of occasions where 'factor of ten' errors occur, and these were corrected as appropriate. It was also noted that across most stations for the period from 1966-1975 there were a lot of SD values of '999' at the beginning

and end of the periods of SC, which means that the value was rejected, or an observation was not made. However, after comparing with more recent years, it appears highly likely that these SD data were originally reported as <0.5 cm (WMO code 997), and they were given an SD value of zero.

To maximise the number of years for which SCO could be calculated, small gaps of missing data were infilled using the

following methodology. To ensure that the estimated SD could not deviate too much from possible likely actual values, infilling was only undertaken if the gap (days) multiplied by the difference in SD (cm) between the days preceding and following the gap was less than or equal to 20. In such cases a linear fit was used to estimate the SD to the nearest cm. As an example, for station 30636, the daily SD is missing for the 30th and 31st December 1939. The SD values on 28th December 1939 and 1st January 1940 were 20 cm and 12 cm, respectively. As 2 × 8 is less than 20, the missing data were infilled and given estimated

SD values of 17 cm and 15 cm for the 30th and 31st December 1939, respectively.

To minimise the impact of temporal changes in the availability of validating data, meteorological stations were only included in the analysis if at least 90% of years with SCO data are available in both the 2003-2022 and 1966-2022 periods. The first of these periods is that over which the ERA5 SD data are most accurate (see Sect. 3.1) and the second is when the observational

SDs are considered homogeneous (e.g., Bulygina et al. 2011). Based on these criteria and the requirement to be located within our definition of Eurasia (35-80°N, 30-180°E), a total of 133 such meteorological stations were chosen. Note that we only examined stations north of 50°N as there are very few former Soviet stations south of this latitude and those that are often have missing SCO data as SC is limited. The locations of the 133 SD observation stations are shown in Figure 2 and further details provided in Table S1.

**2.3 ECMWF fifth generation reanalysis (ERA5)**

ERA5 is the current global reanalysis product produced by ECMWF, with data available from 1940 to present. It is based on the Integrated Forecasting System (IFS) Cy41r2, which was operational in 2016 (Hersbach et al., 2020), and has hourly output of global fields on 137 vertical pressure levels at a spatial resolution of ~31 km. In combination with improved modelling of





surface conditions and more consistent sea ice analyses, this helps it generally better reproduce Arctic climate than its
predecessors, including improved total snowfall (e.g., Wang et al., 2019).

In ERA5, assimilation of both in situ and satellite snow observations is undertaken using a 2D optimal interpolation system as
part of the land data assimilation scheme (LDAS) (Hersbach et al., 2020). From 2004 onwards it also employs information on
SC over the Northern Hemisphere derived from the multi-sensor Interactive Multi-sensor Snow and Ice Mapping System
(IMS) system. This is a 4 km resolution binary product produced by NOAA that combines microwave, visible and infrared
satellite images, and manual analysis input: an SD of 5 cm is assigned to all IMS snow-covered pixels below 1500 m. Mortimer
et al. (2020) and Urraca and Gobron (2023) described a 2003/04 discontinuity in ERA5 SD and snow mass, respectively, which
is associated with the start of the assimilation of IMS data. The authors of the second of these papers also demonstrated two
earlier stepwise improvements in ERA5 SD bias in 1977-1980 and 1991/92: these were linked to the assimilation of the first
satellite products and a wider and more complete series of SD observations in Russia and China (e.g., Clelland et al., 2024),
respectively.

Here, ERA5 daily SD (cm) was computed from the mean daily SD in water equivalent and mean daily snow density, which,
in turn, were calculated as the average of 24 hourly output fields.

## 2.4 Twentieth Century Reanalysis version 3 (20CRv3)

The 20CRv3 reanalysis is the latest version of the NOAA 20CR, described by Compo et al. (2011), with the improvements to
version 3 outlined by Silvinski et al. (2019). It uses the National Centers for Environmental Prediction (NCEP) Global Forecast
System v14.0.1 that was operational in 2017. Daily SD data for the 180 years from 1836-2015 were obtained on a 1° × 1°
latitude/longitude grid. In addition, monthly 1000 hPa geopotential height (hereinafter gpht) data were utilised to develop a
winter AO index (see Sect. 2.5) while 850 hPa gpht fields were employed in further analysis, a pressure level chosen to reflect
surface conditions while being above all but the highest mountain ranges. Although 20CRv3 has 80 ensemble members, the
spread of which provides an internal measure of reanalysis uncertainty, in this study we only utilised data from the ensemble
mean.

This reanalysis only assimilates surface pressure data and does not include an analysis of surface conditions. Key updates in
20CRv3, include an upgraded assimilation scheme, which incorporates a 4D incremental update analysis and uses an adaptive
rather than fixed localisation length for quality control, a higher-resolution forecast model and a larger set of pressure
observations made available through various data-mining projects. Of particular relevance to the current study, this has resulted
in a marked reduction of the negative pressure biases found across much of western Eurasia in earlier versions of 20CR
(Silvinski et al., 2019). Snow is represented as a separate layer on top of the soil layer with independent prognostic thermal
and mass content but liquid water content is not accounted for (Wegmann et al., 2017).





Two previous studies have compared snow parameters in earlier versions of 20CR with observations over Eurasia. Peings et al. (2013) found that, by treating SC simply as a binary dataset, 20CRv2 was able to represent the daily advance of SC in October and November throughout the 20[th] century. Wegmann et al. (2017) evaluated SD in four long-term reanalyses, including two earlier versions of 20CR. While the 20CR reanalyses generally overestimated SD, they did demonstrate a moderate daily correlation (0.6-0.7) with observations across the whole 20[th] century and provided a good representation of both spatial and temporal variability of SD in the second half. In particular, the authors noted relatively high correlations with observed SD in 20CRv2 in October-November.

## 2.5 AO indices

We used two winter (December-January-February, DJF) AO indices to examine the relationship between Eurasian SC and the AO. The first of these is taken from the monthly AO index available at the Climate Prediction Center (CPC). This was obtained by projecting monthly 1000 hPa gpht anomalies poleward of 20°N onto the loading pattern of the AO, which is the leading Empirical Orthogonal Function (EOF) as derived from the NCEP/National Center for Atmospheric Research (NCAR) reanalysis for 1979-2000. These data are only available from 1950 onwards. Therefore, in order to investigate the Eurasian SC-AO relationship prior to this a winter AO was derived from 20CRv3 1000 hPa gpht fields using a similar methodology. This approach is similar to Peings et al. (2013), although they used 20CRv2 sea level pressure.

## 3 Methods

### 3.1 Removing bias and recognising temporal discontinuities in 20CRv3 SC

The first stage of the methodology is to examine temporal changes in the accuracy of the ERA5 reanalysis by comparing the SCO determined from daily SD data, interpolated bilinearly to a station location, against that derived from observations at the 133 meteorological stations for the October-November (ON) period (1940-2022). Figure 3 demonstrates that there is a marked improvement in the bias in ERA5 SCO between 2002 and 2003, after which the significant majority of the station locations have zero bias. Therefore, we consider ERA5 SCO to be 'accurate' for the 20-year period from 2003-2022 (hence the period chosen for Fig. 1).

The timing of the improvement in ERA5 SCO in 2003 is close to but does not exactly match the 2003/04 discontinuity in other ERA5 snow parameters previously described by Mortimer et al. (2020) and Urraca and Gobron (2023), which is associated with the start of the assimilation of IMS data. In addition, temporary improvements in ERA5 SCO bias occur at similar times to the earlier stepwise improvements in ERA5 SD bias in 1977-1980 and 1991/92 noted by Urraca and Gobron (2023). However, in the case of SCO the bias subsequently worsens slightly until the next step-change in accuracy.



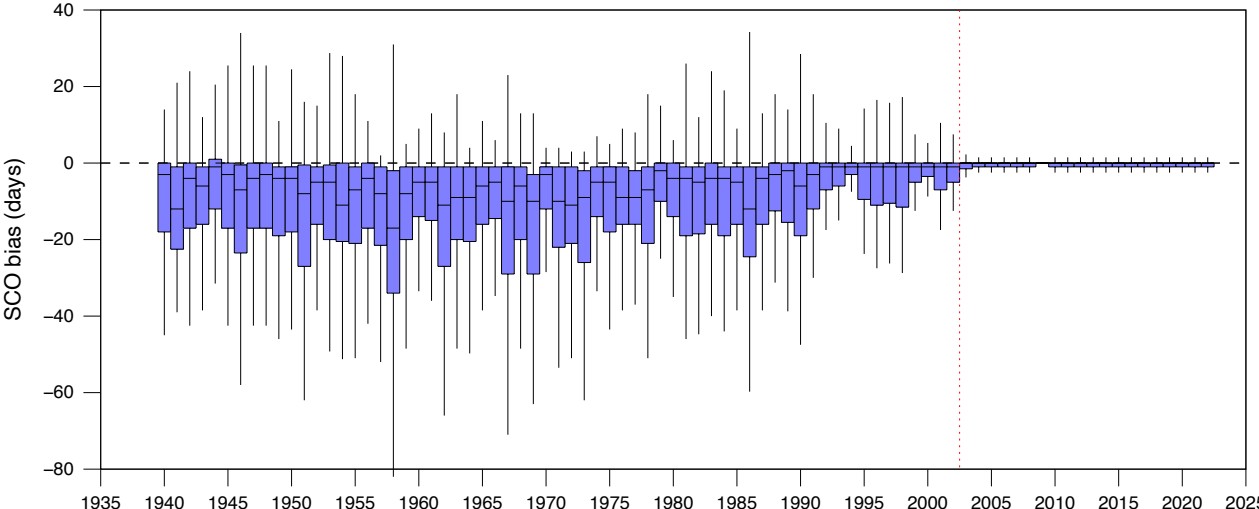

**Figure 3.** Annual bias statistics in SCO in ERA5 (1940-2022). The black horizontal line represents the median bias, the blue boxes show the central 50% range in the bias and the whiskers the larger (smaller) of either the upper (lower) quartile plus (minus) 1.5 times the interquartile range or maximum (minimum) value. The vertical dashed red line indicates when the final discontinuity occurs (2003) after which ERA5 SCO is considered accurate.

SD in earlier versions of 20CR is known to be biased high (Wegmann et al., 2017). Therefore, the second component of the methodology is to calculate any offset in 20CRv3 SD required to minimise the difference in daily SC between that reanalysis and ERA5 for ON during the period of overlap in which the latter SCO is considered accurate (2003-2015). Changes in the bias and RMSE for offsets from 2.5-10.0 cm are shown in Fig. S1, which reveals than an offset of 9.0 cm provides the best statistics. Thus, when calculating the various SC indices in the following analyses 9.0 cm is removed from the 20CRv3 SD. Figure S2 compares the mean daily Eurasian SC in ON between the two reanalyses to examine if there is any marked change in the difference between them across the two months: 20CRv3 has slightly more SC than ERA5 at the beginning of October and end of November and slightly less in the second half of October but these differences are minor.

A comparison of the offset 20CRv3 SCO bias versus that derived from the meteorological stations is shown in Fig. 4. The median bias is generally within one or two days of zero since the early 1920s but there are temporal changes in the distribution of the bias values. For example, while the interquartile range is typically ~10 days from the 1920s onwards the distributions are predominantly negatively skewed with bias values generally negative (SCO too early) until ~1966 after which they become consistently positively skewed with bias values mostly positive (SCO too late). The timing of the switch aligns with both the beginning of homogeneous SD measurements at the stations (Bulygina et al., 2011) and when there is almost 100% data availability of these validation data (cf. Fig. 4). Equivalent analysis of the top 10% of stations with the most years of SCO data



available (not shown) reveals that this improvement is not simply an artifact of the number of stations available for validation. Thus, we consider the SC indices derived from 20CRv3 to be most accurate from 1966 onwards. In addition, the lower and more consistent SCO bias distributions from 1921 onwards suggests that the reanalysis is more accurate in this period than prior to this date, when there is much greater variability in the bias, both between years and between stations. Therefore, in

this analysis, we examine the SC indices across three separate periods; (a) 1836-2015 (the whole range of 20CRv3); (b) 1921-1965; and (c) 1966-2015. Based on Fig. 4, results can be considered to be likely increasingly accurate from (a) to (c): nevertheless, with periods (b) and (c) being of similar length, we also compare changes in the SC indices between these two periods.

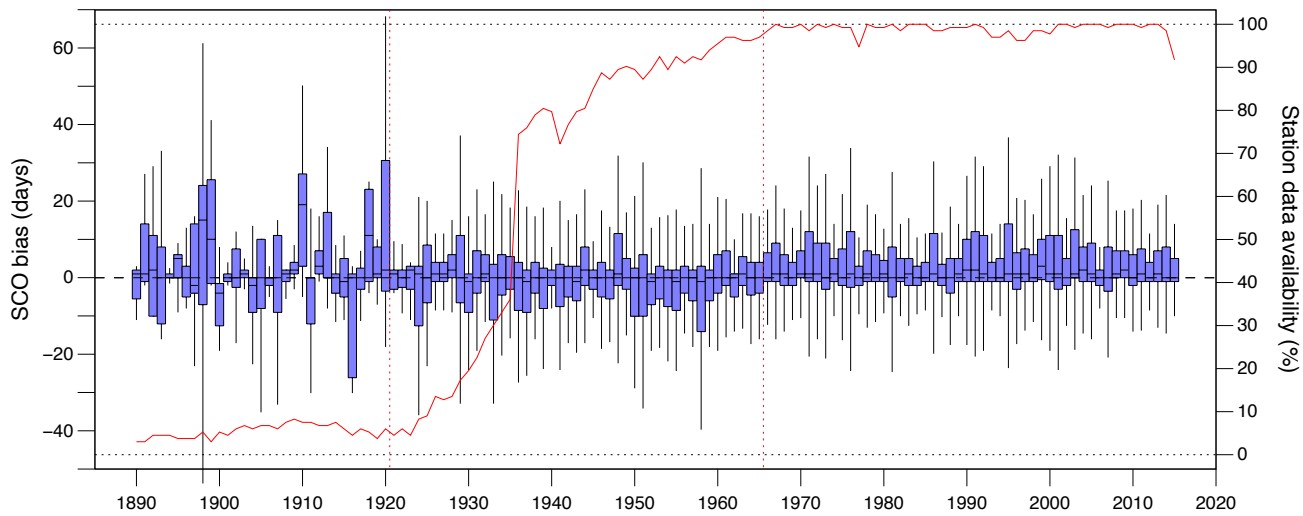


**Figure 4.** As Figure 3 but for SCO in 20CRv3 (1881-2015) with an assumed 90 mm bias in SD removed. The red line is the proportion of the 133 in situ station measurements available for comparison. The vertical dashed red lines are positioned at 1921 and 1966, which represent the start of different periods of likely 20CRv3 SD accuracy (see text).

### 3.2 Statistical methods

Trends in the SC indices were calculated using the non-parametric 'Sen's slope' method (Sen, 1968) that reduces the sensitivity to outliers. Furthermore, we employ the Mann-Kendall test to calculate the significance of the trends because the data may not be normally distributed as there are limits of 0% and 100% for the SC indices. We considered utilising prewhitening of the data to eliminate the effects of any autocorrelation. However, while this method removes the possibility of finding a significant trend in the Mann-Kendall test when there is none, it also has the disadvantage of potentially accepting the null hypothesis of

no trend with a high probability when a trend does exist. Based on the results of the simulation study of Bayazit and Önöz (2007), we decided not to use prewhitening in our analyses as it would reduce the power of the Mann-Kendall test in the



majority of the trends examined. Trends were only calculated when at least 50% of potential data were available: 'missing' reanalysis data result from a region either having 0% or 100% SC on all days in a month and thus no SAI can be calculated. Decadal correlations between the SC indices and the AO were calculated once both datasets had been detrended across that
decade.

## 4 Results

### 4.1 SC trends

Full 20CRv3 time-series of the SCIs, SAI_OCT and SAI_NOV are shown in Figs. 5a, 5b and 5c, respectively, and trends for
the SC indices for the three periods examined given in Table 2. Across the 180 years of reanalysis data there is a small — but nevertheless statistically significant ($p < 0.10$) — increase in mean September Eurasian SC. In contrast, there has been significant decreases in both mean October and December SC ($p < 0.01$). Trends in the two SCAs across the entire 180 years are not significant.

For the 1921-65 period none of the SC indices have significant trends but it is worth noting that all three SCIs have positive trends. SAI_OCT and SAI_NOV have similar magnitude trends of opposite sign. This suggests that a more rapid snow advance in October necessarily contributes to a slower advance in November, as more area is already snow covered at the beginning of the month and vice versa. However, as there is also a contemporaneous increase in SCI_11 other factors such as changes in regional climate are also likely involved.

Interestingly, Fig. 5 and Table 2 reveal that the trends in 1921-65 and 1966-2015 are of opposite sign for all five SC indices. Thus, 20CRv3 demonstrates a decrease in mean SC in all three autumn months for 1966-2015, being statistically significant in October ($p < 0.01$) and November ($p < 0.05$). In these two months the trend magnitude is also much greater than in the preceding period, contributing to the significantly negative trends across the whole span of 20CRv3 (cf. Fig. 5a). The trends
in the SAIs exhibit an increase in SAI_OCT (slower advance) and a decrease in SAI_NOV ($p < 0.01$) (faster advance) over 1966-2015, which corresponds to a greater contemporaneous decrease in SC in October than November (cf. Table 2) and thus an overall postponement of SCO in this time period. Figure 5c shows that the marked decrease in SAI_NOV in 1966-2015 results from a switch from the highest decadal values of the entire 20CRv3 time-series in the 1970s to the lowest in the 2000s. The negative trends in autumn months from 20CRv3 for 1966-2015 match those derived from satellite data for shorter periods
within this 50-years (e.g., Brown and Derksen, 2013; Hori et al., 2017), while the switch in sign of the SC trends across the two recent periods was noted in earlier studies (e.g. Callaghan et al., 2011).

Next, we examine the statistical significance of SCI trends in the sub-regions of Eurasia to evaluate whether there are any distinct spatial patterns of change contributing to the overall trends in Eurasian SC previously discussed. These results are



**Figure 5.** Annual Eurasian SC index data from 20CRv3. (a) SCI_09 (blue), SCI_10 (brown) and SCI_11 (red); (b) SCA_OCT and (c) SCA_NOV: annual data (dotted lines) and running decadal means (full lines) Black lines show trends for 1836-2015, 1921-1965 and 1966-2015.




**Table 2.** Decadal trends of the snow indices derived from 20CRv3 for the three time periods examined. Units for the SCI indices are in $km^2$ $\times 10^6$ $dec^{-1}$ and the SCA indices are in $dec^{-1}$. Asterisks denote trends that are statistically significant: * $p < 0.10$; ** $p < 0.05$; and *** $p < 0.01$.

| Snow Cover Index | 1836-2015 | 1921-1965 | 1966-2015 |
|---|---|---|---|
| SCI_09 | 0.0102* | 0.0795 | –0.0316 |
| SCI_10 | –0.1031*** | 0.1659 | –0.3334*** |
| SCI_11 | –0.0973*** | 0.0858 | -0.2645** |
| SAI_OCT | 0.0127 | –0.0620 | 0.1374 |
| SAI_NOV | 0.0001 | 0.0613 | –0.3024*** |

presented in Fig. 6. Unsurprisingly, given the very limited SC in September (Fig. 1a) there are relatively few sub-regions with enough data to provide 50% of data for trends in SCI_09. The overall increase in this SC index across 1836-2015 appears driven by increases in September SC extent in the region around Lake Baikal and in the far north-east of Eastern Siberia (Fig. 6a). However, we note the first of these regions does not demonstrate a significant trend in either of the two shorter periods, while in the latter (1966-2015) there have been reductions in SC in northern Eurasia, north of 70°N (Fig. 6c).

Figure 6 reveals that trends in SCI_10 and SCI_11 have broadly similar spatial patterns in significant trends to each other across all three periods examined. For 1836-2015 there is a distinct latitudinal division, with statistically significant trends towards less SC in most of the subregions in the north and more SC in the some of the subregions in the south (Figs. 6d and 6g). In 1921-1965 all the subregions with significant trends in SCI_10 and SCI_11 (with one exception) have positive trends. For SCI_10 there is no clear spatial pattern to these areas whereas for SCI_11 they are limited to being west of 90°E (Figs. 6e and 6h). Comparison with Figs. 1a and 1b indicates that the majority of subregions with these trends are located beyond the limit of the monthly median SC for 2003-2022, suggesting that the spatial pattern of the trends is primarily associated with a southern expansion of autumn SC in western Eurasia during this period. For the most recent period (1966-2015) the subregions having statistically significant negative trends in SCI_10 and SCI_11 are more spatially coherent, being confined to northern Eurasia, similar to SCI_09, but also extending south to 50°N in the west of the region (Figs, 6f and 6i), a pattern which, is broadly equivalent to the trends described by Bulygina et al. (2009) for 1966-2007.

Both SAIs have many subregions with significant trends across 1836-2015 (Figs. 6j and 6m). However, as there are broadly similar numbers of subregions with positive and negative trends, there is no trend for Eurasia as a whole for either SC index (cf. Table 2). There is some spatial coherence to the distribution of subregions with a significant trend of one sign but it is markedly less distinct than for SCI_10 and SCI_11. For SAI_OCT the general pattern is for positive trends (slower snow





advance) to the west and negative trends (faster snow advance) to the east, whereas for SAI_NOV it is predominantly positive (negative) trends in the south and west (north and east) of Eurasia.

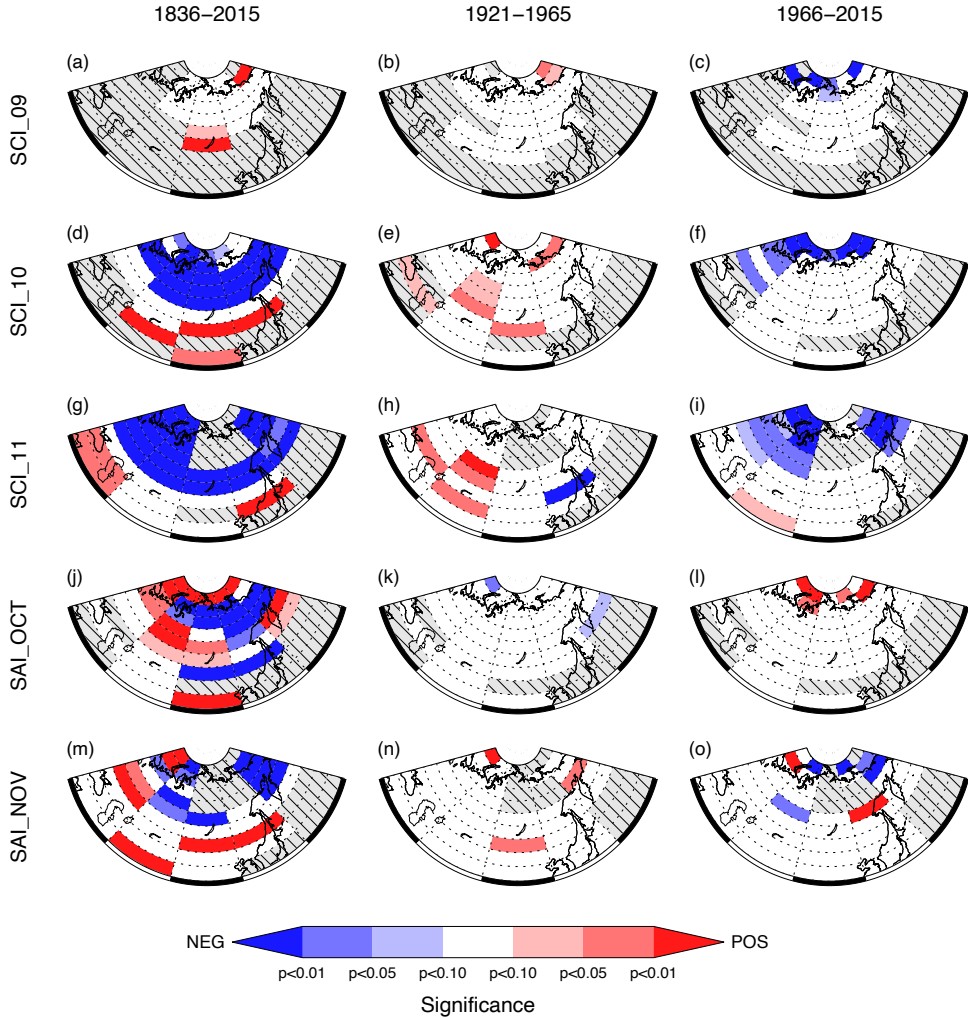

**Figure 6.** Statistical significance of trends in SC indices: SCI_09 (row 1), SCI_10 (row 2), SCI_11 (row 3), SAI_OCT (row 4) and SAI_NOV (row 5) for 1836-2015 (first column), 1921-1965 (second column) and 1966-2015 (third column). Hatched subregions are where no value was calculated because data availability was less than 50%.

During the two shorter periods examined there are only a few subregions with statistically significant trends in the monthly SAIs and these are primarily located in the north of Eurasia. In the 1921-1965 period they are solely negative (2) for SAI_OCT and positive (3) for SAI_NOV (Figs. 6k and 6n, respectively). The sign of these trends matches that of the SAI trends for





Eurasia as a whole, which are not significant (cf. Table 2). In 1966-2015 the sign of all the significant trends switches from negative to positive for SAI_OCT (Fig. 6l). These trends are confined to the highest latitudes of Eurasia and include a switch

in sign for the subregion 30-60°E, 75-80°N, such that during the earlier period the rate of snow advance in October accelerated and subsequently decelerated. The majority of significant SAI_NOV trends in 1966-2015 are negative, unsurprising given the highly significant trend for all of Eurasia, but nevertheless there are also two subregions with a significant positive trend (Fig. 6o). Similar to SAI_OCT, the majority of subregions with significant trends are found in the north of Eurasia.

Across all three periods there are several examples when a subregion has significant trends in SAI_OCT and SAI_NOV of opposite sign, comparable to Eurasia as a whole. However, comparison of Figs. 6j and 6m indicates that this is not always the case and indeed there can be significant trends of the same sign in both periods (e.g., 70-80°N, 30-60°E).

**4.2 SC relationship with the AO**

Time series of decadal correlation coefficients ($r$) between the 20CRV3 SC indices and the two AO indices — from 20CRV3
(1831-2015) and CPC (1950-2015) — are illustrated in Fig. 7. Equivalent data derived from ERA5 (2003 onwards) and the CPC AO are also shown in order to bring the time-series of correlations up to date. Fig. 7 indicates little difference in the $r$ values between the two AO indices for the period of overlap. Differences between the AO correlations using 20CRv3 and ERA5 SC indices are also small, with the exception of SCA_NOV: Fig. 7e reveals that the SCA_NOV-AO correlation for 2004-2013 was –0.42 (not significant) for 20CRV3-CPC AO and –0.73 ($p < 0.05$) for ERA5-CPC AO. Table 3 summarises
the proportion of decades with a positive or negative SC_AO relationship and those that are statistically significant ($p < 0.10$).

Figures 7a and 7b indicate broad similarities in the temporal variability of the SCI_09-AO and SCI_10-AO relationships. Both SC indices have a mix of decades with positive and negative relationships with the AO, with the latter more prevalent, especially for SCI_09 (Table 3). Both SC indices also demonstrate a distinct trend from negative to positive values from the
1970s to recent decades. The first half of this trend in SCI_10 can be seen in the earlier work of Peings et al. (2013; cf. their Fig. 3b). There are no decades with a statistically significant positive relationship between SCI_09 and the AO, although we note that the highest correlations occurred in recent decades (Fig. 7a). Short periods with significant negative correlations happened in the 1860s and 1920s, with a longer period in the 1940s. For SCI_10, the period with the most consistent significant negative correlations with the AO was in the 1970s, at the start of the recent positive trend, with a period of smaller but still
significant correlations in the 1930s. There were no periods with a significant negative correlation before this time (Fig. 7b). In contrast, the only period with a significant positive trend was centred around the 1850s and the recent values derived from ERA5 are the highest since then. Both SCI_09 and SCI_10 have more than 10% decades with a statistically significant negative relationship with the AO, which is greater than for any other SC index examined here (cf. Table 3).




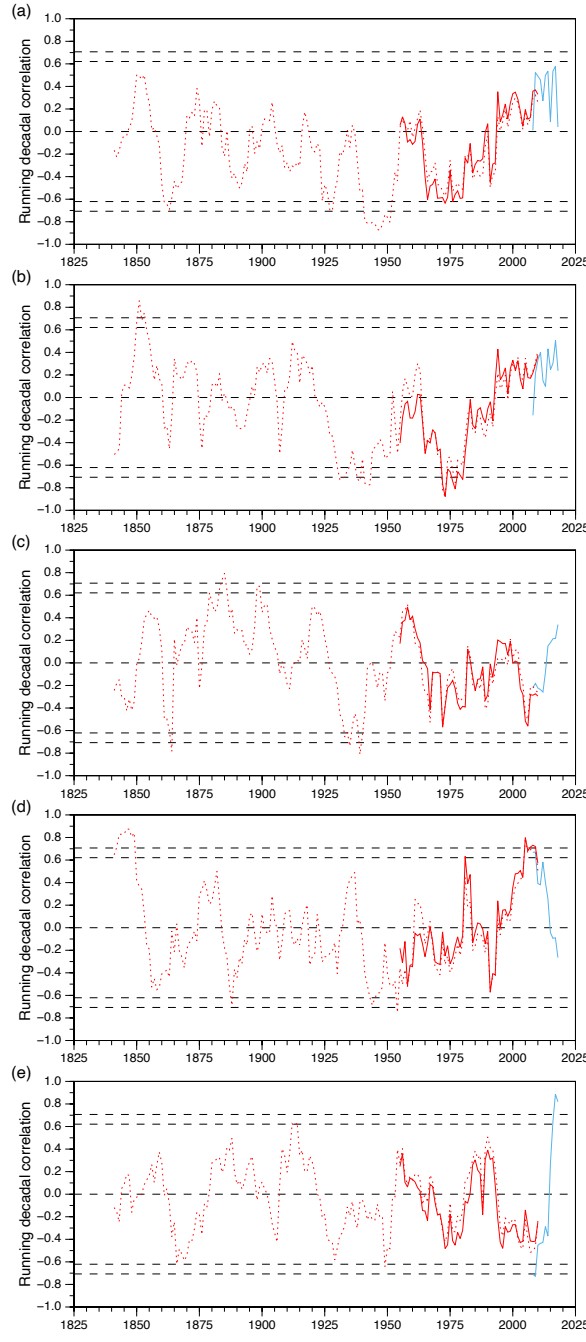

**Figure 7.** Running decadal correlations between snow cover indices and winter AO: (a) SCI_09; (b) SCI_10; (c) SCI_11; (d) SAI_OCT; (e) SAI_NOV. Snow cover indices from 20CRv3 are shown and ERA5 (2003-12 to 2012-21) as red and blue lines, respectively. Correlations with winter AO from CPC and derived from 20CRV3 are shown as full and dashed lines, respectively. The black dashed horizontal lines indicate significance levels at $p < 0.10$ and $p < 0.05$ for each decade considered independently.





**Table 3.** Proportion of decades that the 20CRv3 SC indices have positive and negative correlations with the 20CRv3 AO, and the proportion that are statistically significant ($p < 0.10$). The period covered is 1831-40 to 2006-2015


| Snow Cover Index | Positive (%) | Negative (%) | Pos. (significant) (%) | Neg. (significant) (%) |
|---|---|---|---|---|
| SCI_09 | 37.1 | 62.9 | 0.0 | 11.8 |
| SCI_10 | 47.6 | 52.4 | 2.9 | 11.2 |
| SCI_11 | 50.6 | 49.4 | 4.1 | 5.3 |
| SAI_OCT | 38.8 | 61.2 | 8.2 | 5.3 |
| SAI_NOV | 45.3 | 54.7 | 1.8 | 2.4 |

The decadal correlation values of SCI_11-AO are more evenly distributed between positive and negative than the SCI indices of the earlier months (Table 3) although negative values are most frequent for recent decades and there are still more decades with significant negative correlations than positive. However, there have not been any statistically significant decadal

correlations since the 1930s; that is, there are none observed for the more reliable recent 20CRv3 data. Prior to that were short periods of significant correlations in the 1860s (negative), 1880s (positive) and 1890s (positive).

The running decadal correlations between the SAI_OCT and AO and SAI_NOV and AO are shown in Figs. 7d and 7e, respectively. In the former there are significantly more negative than positive values (Table 3), suggesting that the period of

strong positive correlations that occurred in the 1990s and 2000s, as described by Cohen and Jones (2011), was a rather exceptional event, as also previously noted by Peings et al. (2013). Nevertheless, there are more decades with significant positive than negative relationships between SAI_OCT and the AO. In addition to the recent period in the 2000s, Fig. 7d reveals a period of positive correlations centred in the 1840s, which precedes, rather than coincides with, the period of significant positive SCI_10-AO correlations (Fig. 7b). There are three periods of significant negative correlations in the

SAI_OCT-AO time series, which are shorter and less statistically significant than the equivalent positive periods. Finally, the ERA5 data reveal that the recent period of statistically significant relationship between SAI_OCT and the AO has ended: the most recent decades have negative $r$ values.

Table 3 reveals that there are far fewer decades with a significant relationship between SAI_NOV and AO than for the other

SC indices. As the frequencies of both significant positive and negative decades are <5% it could be argued that these relationships are simply stochastic noise. In the 20CRv3 data there is only one period each of significant positive and negative correlations. However, in stark contrast the ERA5 data indicate a very fast change from statistically significant negative values





(–0.73 for 2004-13) to significant positive values (0.89 for 2012-21). As the ERA5 SAI_NOV-AO correlations immediately prior to 2003-1012 (not shown) are broadly similar to those from 20CRv3, and with the caveat that the ERA5 SC data are not

temporally homogeneous, this implies a stronger and perhaps more variable relationship between SAI_NOV and winter AO i

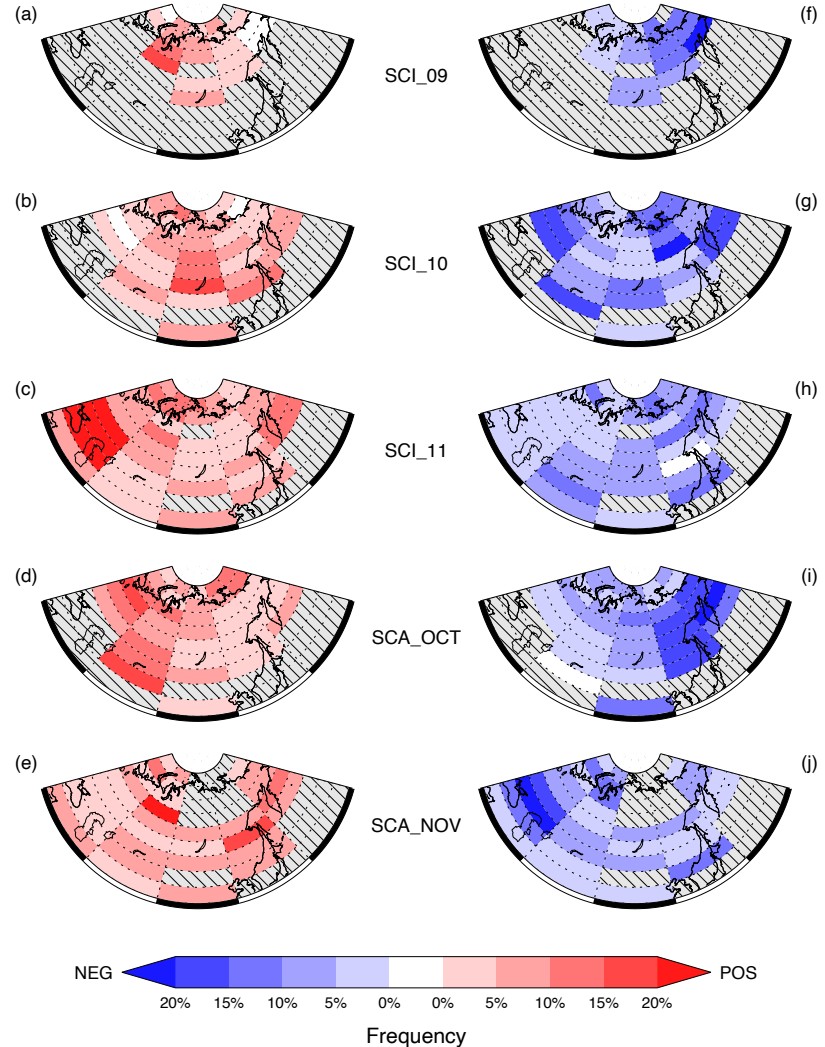

**Figure 8.** The frequency of decades with statistically significant correlations with the AO ($p < 0.10$) from the 20CRv3 data (1831-1840 to 2005-2014) for the five SC indices. Frequencies for positive correlations are shown on the left in red and negative correlations on the right in blue. Hatched subregions are where no value was calculated because data availability was less than 50%.






in recent decades. Likewise, Yang and Fan (2021) observed a strengthened relationship between November Siberian SC and the winter Siberian High and related it to changes in troposphere-stratosphere coupling and associated changes in the AO.


Next, we investigate the spatial variability in the frequency of significant between the SC indices and the AO across Eurasia (Fig. 8). Interestingly, there are very few subregions that have decades with a significant relationship with the winter AO of only one sign (e.g., those shaded white in Fig. 8) and there are none for SAI_NOV.

The SCI_09 data indicate that within the northerly subregions, where there are sufficient decades with SCI_09 correlations to provide a reliable value (≥ 50%), there is a general pattern of more significant positive decadal correlations with the AO in the west and more significant negative correlations in the east of Eurasia, with frequencies exceeding 10% east of 120°E (Figs. 8a and 8b). In contrast, the SCI_10 data show less spatial coherence in the pattern of frequencies. The subregions with the highest proportion of significant positive relationships with the AO tend to be more in the centre of Eurasia while those with a greater

fraction of significant negative relationships are more peripheral, at the edge of the expanding SC, with frequencies of more than 15% in both west and east (Figs. 8c and 8d). There is a statistically significant relationship between SCI_11 and the AO more than 20% of the time in the south-east of Eurasia (40-55°N and 30-60°E) (Fig, 8e), predominantly located beyond the median November SC (cf. Fig. 1c). There is less spatial coherence across the rest of Eurasia although there is a propensity for subregions with higher frequencies of significant positive correlations to have coastal locations. The equivalent plot for

significant negative relationships (Fig. 8f) has very low frequencies in the west of Eurasia: less than 5% for most of the region west of 90°E. Frequencies further east are higher but no subregion has more than 15%.

The SAI_OCT data reveal a broadly longitudinal split in the sign of subregions with significant decadal relationships with the AO (Figs, 8g and 8h). Clearly, significant positive relationships between SAI_OCT and the AO are more frequent in western

Eurasia while significant negative relationships are more common in the east, with a frequency >15% in the majority of sub-regions east of 120°E. The fact that frequencies for significant negative (positive) SAI_OCT-AO relationships in the west (east) are predominantly less than 5% adds further weight to the importance of this longitudinal influence of Eurasian SC on the following winter AO. This finding is analysed further in the discussion (Sect. 5). The equivalent pair of figures for SAI_NOV (Figs. 8i and 8j) do not show the same kind of clear spatial pattern in frequencies. If anything, the patterns are

somewhat reversed, with higher frequencies of significant positive SAI_NOV-AO relationships in the east and greater frequencies of significant negative relationships in south-western Eurasia, where values exceed 20% in the 45-50°N, 30-60°E sub-region.

Finally, for the decades with the strongest correlations between the SA indices across the whole of Eurasia and the AO, we

examine the extent to which the subregions contribute to the overall SAI-AO relationships. For consistency we only include periods encompassed by the 20CRv3 data. Thus, we analyse the strongest positive SAI_OCT-AO correlation that occurred in



2000-09 ($r = 0.79$) and the equivalent strongest negative correlation in 1949-58 ($r = –0.76$) (cf. Fig. 7d). The strongest positive and negative relationships for SAI_NOV-AO are in 1985-94 ($r = 0.51$) and 1944-53 ($r = –0.65$), respectively (cf. Fig. 7e).

Figure 9a shows the SAI_OCT-AO correlations across the subregions for the strongly positive decade. Clearly, across the majority of Eurasia there is a positive relationship between SAI_OCT and the AO but nevertheless there are subregions with strongly negative correlations in the north and south of Eurasia (Fig. 9a). We note that the subregions with the strongest positive

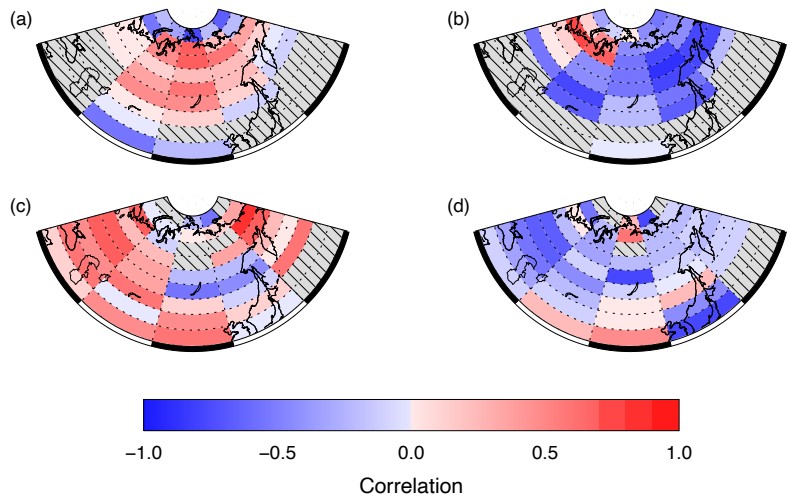

**Figure 9.** Spatial variability in the correlation between SAI_OCT and AO for decades when the relationship is (a) most strongly positive
and (b) most strongly negative for Eurasian SC as a whole. (c) and (d) are the same for SAI_NOV. Hatched subregions are where no value was calculated because data availability was less than 50%.

correlations lie between 90-120°E, which is not the region with the highest frequency of statistically significant positive correlations (cf. Fig. 8g). This indicates that when a period with an especially strong SAI_OCT-AO relationship occurs it is at least partially a result of a greater area of Eurasia contributing to the sign of that relationship rather than simply a much stronger relationship for those subregions that usually have a significant relationship of that sign with the AO. A comparable event happens in the decade of strongly negative SAI_OCT-AO relationship (Fig. 9b). Strong negative correlations in eastern Eurasia

occur as is often the case (cf. Fig. 8g) but the region with negative correlations also extends westward across most of Eurasia, including several sub-regions where significant positive SAI_OCT-AO relationships are more frequent than the negative equivalents (e.g. 60-90°E).





The *r* value for the decade with the strongest positive SAI_NOV-AO decade is not statistically significant, and this may be
why there is a less consistent pattern of correlation values across the subregions than for the other examples in Fig. 9. Figure
9c has a mix of positive and negative correlations, with a broadly south-north split, with the strongest positive correlations
contributing to the overall SAI_NOV-AO relationship located at 40-55°N, 60-90°E. All but one subregions east of 120°E
actually have a negative relationship with the AO in this decade. The decade with the strongest negative SAI_NOV-AO
correlation has a somewhat similar pattern to that of the frequency of significant negative correlations (cf. Figs. 9d and 8j).
The negative correlations of greatest magnitude are found in western Eurasia but, unlike the frequency plot, the highest values
are actually located north of 60°N, between 60-90°E. Many of the subregions in the east of Eurasia have a positive correlation
with the AO in this decade, that is, similar to the positive SAI_NOV-AO example, they are of opposite sign to the overall
Eurasian SAI_NOV-AO relationship.

**5 Discussion**


One of the most noteworthy results from the analysis of the SC-AO relationships is the distinct longitudinal dipole in the sign
of the frequency of decades with statistically significant correlations between SAI_OCT and the following winter AO (cf.,
Figs. 8g and 8h). To examine this relationship further, separate running decadal SAI_OCT-AO correlations for east and west
regions of Eurasia are shown in Fig. 10, where east is defined as 45–70°N, 120–180°E and west as 45–70°N, 30–90°E. The
correlation between the two time-series of SAI_OCT correlations is –0.29, which, although not quite statistically significant
at $p < 0.10$, does confirm the general negative relationship between SAI_OCT-AO in the east and west of Eurasia suggested
by Fig. 8. That is, typically, the rates of October snow advance in the two regions have oppositely signed connections with the
following winter AO.

In Fig. 10 we also plot the sum of the east and west SAI_OCT-AO correlations, which very closely matches the SAI_OCT-
AO data for the whole of Eurasia (Fig. 7d; $r = 0.84$, $p < 0.01$). Thus, the periods when the SAI_OCT-AO relationship is
statistically significant for Eurasia as a whole generally occur when both east and west regions have an SAI_OCT-AO
correlation of the same sign, which corresponds to the findings of the analysis of the decades with the strongest relationships
(Figs. 9a and 9b).  For example, the recent period when there was a positive SAI_OCT-AO, as described in Cohen and Jones
(2011), is primarily due to a switch in sign of the SAI_OCT-AO relationship in eastern Eurasia from negative to positive. That
means that, atypically, during this time a more rapid (slower) advance in October SC in eastern Eurasia was associated with a
negative (positive) AO in the subsequent winter.

Next, we undertake a brief analysis to investigate whether there are any key differences in the lower atmospheric circulation
between the 'recent period' (2000-2009 to 2005-2014), with the statistically significant positive SAI_OCT-AO correlation,
and the prior period when no significant SAI_OCT-AO relationship existed. We focus on the period from 1966 onwards, when
the 20CRv3 SC values are likely more accurate and homogeneous (cf. Fig. 4). Figure 11a shows the climatological October



850 hPa gpht field, which generally exhibits low pressure over the ocean and high pressure over land, the latter dominated by the developing Siberian High centred south of the Central Siberian Plateau over northern Mongolia. The anomalous decadal

October 850 hPa gpht field for the recent period is displayed in Fig. 11b and reveals two principal areas of gpht difference. There is anomalously high pressure in the vicinity of the Ural Mountains (~60°E) and low pressure over the Bering Sea. The former describes an increase in Ural Blocking (UB), which is itself a source of Eurasian climate predictability and a precursor to a weakened stratospheric vortex. Indeed, some previous work suggests that the UB-AO relationship is more robust than that for SC-AO (e.g., Peings, 2019). The anomalously negative pressure over the Bering Sea could potentially be associated with

the positive polarity of the Pacific North American (PNA) pattern of atmospheric circulation variability, which was indeed more positive on average in the recent period than before, and pressure in this region has previously been linked to Eurasian SC (Garfinkel et al., 2010). Elsewhere, there is a slightly weakened Siberian High (lower gpht), especially in the region around

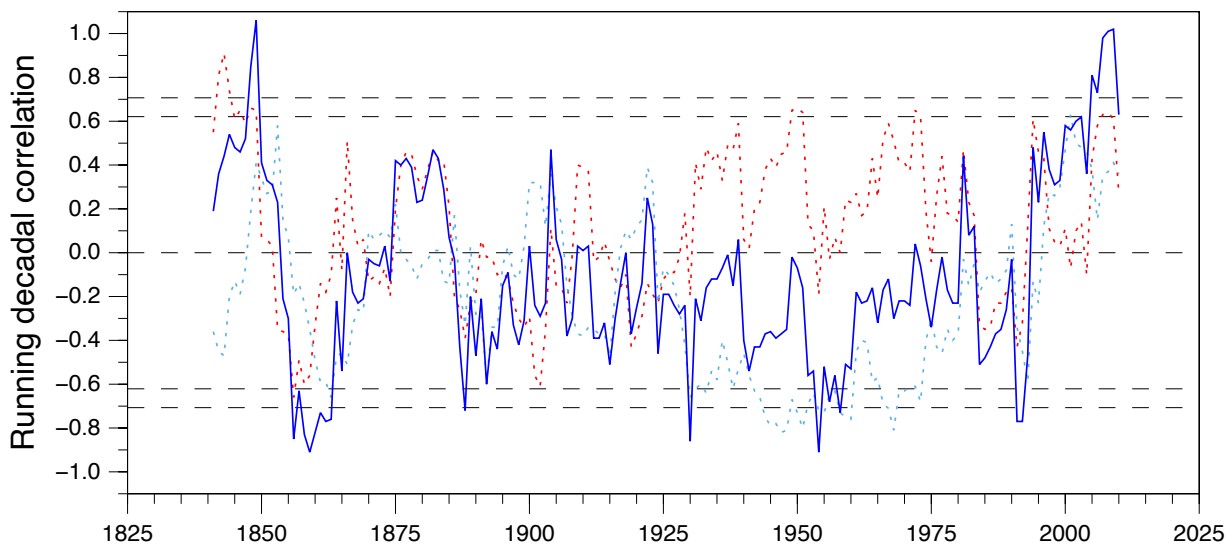


**Figure 10.** Running decadal correlations between SAI_OCT and winter AO derived from 20CRv3. SAI_OCT from west and east Eurasia are shown as red and blue dotted lines, respectively and the full dark blue line is the sum of the correlations from the two regions. The black dashed horizontal lines indicate significance levels at $p < 0.10$ and $p < 0.05$.

Lake Baikal and reduced gpht in the Barents-Kara Sea. The latter may be associated with changes in sea ice: reduced regional sea ice cover has also been proposed as a driver of anticyclonic ridging over the Urals with a consequent impact on vertical wave propagation and the AO (e.g. Cohen et al., 2014b).

Figures 11c and 11d show the correlation between decadal October 850 hPa gpht and SAI_OCT-AO correlations in west and

east Eurasia, respectively. Both regional SAI_OCT-AO correlation time-series have a negative correlation in the Lake Baikal



region, indicating that a faster October SC advance (negative SAI_OCT) in either region leads to a stronger Siberian High due to enhanced diabatic cooling (Cohen et al., 2007). In Fig. 11c there are negative correlations across most of western Eurasia and positive correlations east of ~130°. Interestingly, while the correlation with 850 hPa gpht associated with UB is clearly evident as being less negative (or indeed slightly positive) compared to the surrounding region, its magnitude is close to zero.

This contrasts with Fig. 11d, which demonstrates that a positive SAI_OCT-AO for SC in east Eurasia is linked with greater gpht in the vicinity and downstream of the Urals. The region of strong negative correlations east of Lake Baikal broadly corresponds to the negative gpht anomaly in Fig. 11b. Crucially, this dipole of pressure anomalies in Fig. 11d is known to be a spatial configuration that provides enhanced vertical propagation of Rossby waves into the stratosphere (e.g., Cohen et al., 2014a) and thus influences the AO. As the dipole is less distinct in Fig. 11c, this suggests that October SC advance variability

in eastern Eurasia was primarily the cause of the significant positive SAI_OCT-AO relationship in the recent period, a conclusion that corresponds to the fact that the contemporaneous sign of the regional relationship was anomalous.

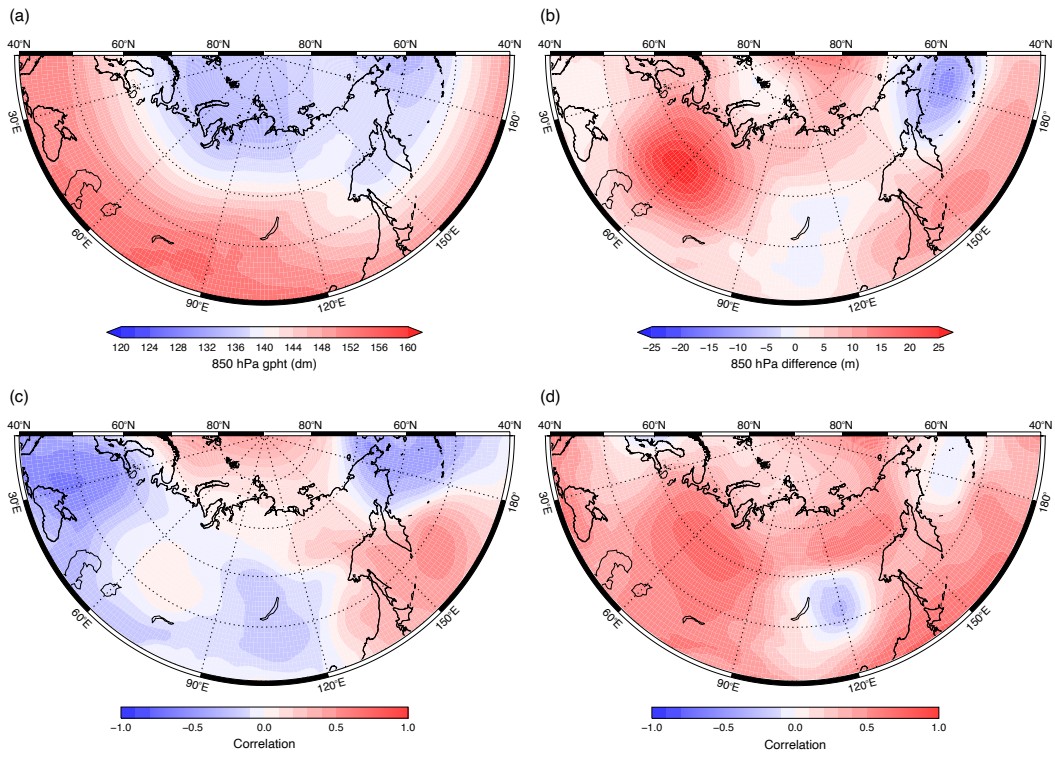

**Figure 11.** (a) Climatology of October 850 hPa gpht from 20CRv3 (1966–2015); (b) Difference in decadal October 850 hPa gpht (2000-2009 to 2005-2014 minus 1966-1975 to 1999-2008); (c) Correlation between decadal SAI_OCT-AO correlations for western Eurasia and October 850 hPa gpht; (d) As (c) but for eastern Eurasia.





**6 Conclusions**


In this study we have investigated spatial and temporal changes in autumn Eurasian SC and its relationship with the following winter AO. We utilised the 20CRv3 reanalysis, which allowed our analyses to go back as far as 1836. As previous studies demonstrated a marked overestimation of SD in earlier versions of 20CR (Wegmann et al., 2017), we calibrated 20CRv3 SC extent against ERA5 data, which in turn were validated against SCO data derived from SD observations. An offset of –9.0 cm

provided the most similar 20CRv3 SC values to ERA5. Comparison of 20CRv3 SCO against observations allowed a qualitative analysis of the temporal homogeneity and likely accuracy of the reanalysis SC data. Lower and more consistent SCO bias distributions from 1921 onwards suggest that the reanalysis is more accurate in this period than prior to this date. However, the most accurate and consistent data is post 1966, which aligns with the beginning of the period of homogeneous SD measurements at the stations (Bulygina et al., 2011).


We examined five SC indices — three monthly SC extent indices (SCIs) and two monthly SC advance indices (SAIs) — across Eurasia as a whole and also as separate 5° latitude by 30° longitude subregions. We considered trends in these SC indices across three time periods: the full 20CRv3 time period (1836-2015) and two shorter periods with likely improved accuracies of the 20CRv3 SC data, 1921-1965 and 1966-2015. Across the full 180 years there is a small — but nevertheless statistically

significant ($p < 0.10$) — increase in mean September Eurasian SC. In contrast, there has been significant decreases in both mean October and December SC ($p < 0.01$). Trends in the two SCA indices across the entire 180 years are not significant.

For 1921-65 none of the SC indices have significant trends. Trends in 1966-2015 are of opposite sign to 1921-65 for all five SC indices, with the later period having a decrease in mean SC in all three autumn months, statistically significant in October

and November. In these two months the magnitude of the trends is greater than in the preceding period, contributing to the significantly negative SC trends across 20CRv3 as a whole. The trends in the SAI indices demonstrate a slowing of October snow advance and an accelerating snow advance in November ($p < 0.01$) over the past 50 years, corresponding to a greater contemporaneous decrease in SC extent in October than November and thus an overall postponement of SCO in this time period.


SCI_10 and SCI_11 have broadly similar spatial patterns in significant trends across the subregions for all three periods examined. For 1836-2015 there is a distinct north-south division, with trends towards less SC in the north and more SC in some of the subregions in the south. In 1921-1965 the subregions with significant trends in SCI_10 and SCI_11 have positive trends, primarily associated with a southern expansion of autumn SC in western Eurasia. For 1966-2015 the subregions with

statistically significant negative trends in SCI_10 and SCI_11 are more spatially coherent, being predominantly confined to northern Eurasia.



The SAI indices have many subregions with significant trends across 1836-2015 but, as there are broadly similar numbers of positive and negative trends, there is no significant trend for Eurasia as a whole. In October the general pattern is faster (slower) snow advance to the east (west) of Eurasia, whereas in November it is predominantly faster (slower) trends in the north and east (south and west). During the two shorter periods examined the few subregions with statistically significant trends in the monthly SAI indices are primarily located in north Eurasia.

Across all three time periods there are several examples when a subregion has significant trends in SAI_10 and SAI_11 of opposite sign, suggesting that a more rapid (slower) snow advance in October necessarily contributes to a slower (more rapid) advance in November, as more (less) area is snow covered than normal at the beginning of November. However, this is not always the case and indeed there can be significant trends of the same sign in both periods, which points to an expansion or contraction of SC in that subregion.

With regards to relationships between the Eurasian SC indices and the AO, there are general similarities in the temporal variability of the SCI_09-AO and SCI_10-AO relationships: both indices have a mix of decades with positive and negative relationships with the AO, with the latter more prevalent (>10%). Both SCI indices also demonstrate a trend from negative to positive values from the 1970s to recent decades. The decadal correlation values of SCI_11-AO are more evenly distributed between positive and negative although the latter are most frequent for recent decades and there are still more decades with significant negative correlations than positive.

The running decadal SAI_OCT-AO correlations have markedly more negative than positive values, suggesting that the period of strong positive correlations in the 1990s and 2000s, as described by Cohen and Jones (2011), was a rather exceptional event. Nevertheless, there are more decades with significant positive than negative relationships between SAI_OCT and the AO. Up-to-date ERA5 data reveal that the period with a statistically significant positive relationship between SAI_OCT and the AO has ended. There are far fewer decades with a significant relationship between SAI_NOV and AO than for the other SC indices in the 20CRv3 data. However, ERA5 indicates a very fast change from statistically significant negative values (–0.73 for 2004-13) to significant positive values (0.89 for 2012-21), implying a more changeable relationship between SAI_NOV and winter AO in recent decades.

Almost all the subregions have decades with statistically significant positive and negative relationships between the SC indices and the AO. The SCI_09 data have a tendency for more significant positive (negative) decadal correlations with the AO in the west (east) of Eurasia. In contrast, the SCI_10 data show less spatial coherence: significant positive relationships with the AO tend to be located in central Eurasia while those with a greater fraction of significant negative relationships are more peripheral, at the edge of the expanding SC. Of note, in south-east Eurasia (40-55°N and 30-60°E) there is a significant positive relationship between SCI_11 and the AO more than 20% of the time, chiefly located beyond the median November SC.





The SAI_OCT data reveal a broadly longitudinal divide between subregions having higher frequencies of significant decadal relationships with the AO of one sign than the other. Significant positive relationships between SAI_OCT and the AO are more frequent in western Eurasia while significant negative relationships are more common in the east, with the majority of sub-regions east of 120°E having a frequency >15%. The equivalent data for SAI_NOV do not show a similarly clear spatial pattern in frequencies.

For the decade with the strongest positive SAI-AO relationship there is a positive relationship between SAI_OCT and the AO across most of Eurasia but nevertheless there are subregions with strongly negative correlations in the north and south. The subregions with the strongest positive correlations (90-120°E) are not those with the highest frequency of statistically significant correlations, indicating that the high magnitude of the correlation is at least partially a result of a greater area of Eurasia having a positive SAI_OCT-AO relationship than normal rather than simply a much stronger relationship for the subregions that usually have that sign relationship. A comparable pattern happens in the decade of strongest negative SAI_OCT-AO correlation. Equivalent patterns for the decades with the strongest SAI_NOV-AO relationships are less spatially consistent, with a greater proportion of subregions having an opposite sign SAI_NOV-AO relationship to that for Eurasia as a whole.

Our key finding is to have demonstrated for the first time that, typically, the rates of October snow advance in the west and east of Eurasia have opposite effects on the sign of the following winter AO, and that when there is an especially strong SAI_OCT-AO relationship it is because the sign of this relationship in one of the regions is reversed. Future work will aim to determine whether this observation relates to the strength of the November SC dipole (Gastineau et al., 2017; Zhang et al., 2023) and the exact mechanisms and linkages that occur in the context of contemporaneous changes in regional atmospheric circulation and the many additional factors that have been found to influence the SC-AO relationship. Furthermore, we will also investigate whether this finding can be utilised to improve seasonal forecasts of the winter AO.

*Data Availability.* All the datasets employed in this study are publicly available and were accessed during 2023/2024.
Snow depth observations:
1. Historical Soviet Daily Snow Depth (HSDSD), Version 2, https://doi.org/10.7265/N5JW8BS3
2. Met Office Integrated Data Archive System (MIDAS) Land and Marine Surface Stations Data, https://catalogue.ceda.ac.uk/uuid/220a65615218d5c9cc9e4785a3234bd0
3. Ogimet, https://www.ogimet.com
Reanalysis data:
1. ERA5, https://cds.climate.copernicus.eu/cdsapp#!/dataset/reanalysis-era5-single-levels?tab=overview
2. 20CRV3, https://www.psl.noaa.gov/data/gridded/data.20thC_ReanV3.html





AO Index:

  CPC AO data,

  https://www.cpc.ncep.noaa.gov/products/precip/CWlink/daily_ao_index/monthly.ao.index.b50.current.ascii.table

715 *Competing interests.* The contact author declares no competing interests

*Acknowledgements.* This study was funded by the UK Natural Environment Research Council (NERC) Climate Change in the Arctic and North Atlantic Region and Impacts on the UK (CANARI) project (NE/W004984/1).

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
