# Peer review of "An examination of changes in autumn Eurasian snow cover and its relationship with the winter Arctic Oscillation using 20th Century Reanalysis version 3"

_EGUsphere, 2024_

## Author Comment (AC1)

**Reviewer 1**

I thank the reviewer for their constructive comments and especially for the excellent suggestion of using a series of scrambled AO time-series in order to test the statistical significance of the frequency of decades with significant SC-AO correlations.

**GENERAL COMMENTS**

This study first uses the National Oceanographic and Atmospheric Administration (NOAA) 20th Century Reanalysis version 3 (20CRv3), together with some other observational data sets, to investigate trends in the Eurasian autumn (September to November) snow cover in years 1836-2015. Then, the variations in the interannual correlation between Eurasian autumn snow cover and the following winter Arctic Oscillation (AO) index are studied. The study adds to previous knowledge by (i) extending its analysis further back in time than has been done thus far and (ii) by also looking in some detail at the regional characteristics of the trends and the snow cover – AO relationship. In addition to the monthly mean snow cover extent in September, October and November, the rate of advance of snow cover during October and November is also analysed.

The main findings of the study include (i) an overall decreasing trend in snow cover extent in October and November since 1836, but particularly since 1966; (ii) a tendency for anticorrelated variations in the rate of snow cover advance between October and November, which is also reflected in the trends for 1966-2015 (slower advance in October but faster in November, due to a delay in the onset of the snow season); (iii) highly variable decadal correlations between the studied snow cover indices and the winter AO; and (iv) the rates of October snow advance in western and eastern Eurasia have typically opposite correlations with the subsequent winter AO index.

One may question the meaningfulness of analysing snow cover variations back to the year 1836, when little data was available to constrain the 20CRv3 reanalysis. However, I believe that this analysis still makes sense, particularly for the interannual variations. Assimilating surface pressure observations, the reanalysis should at least qualitatively capture the snow cover variability, which is largely forced by atmospheric circulation on the interannual time scale. For the same reason, the reanalysis also likely qualitatively captures the correlation between the autumn snow cover and the winter AO even in the 19th century, even though the number of pressure observations was much smaller than today.

My main concerns about this study relate to Section 4.2. Specifically, the manuscript suffers from a too uncritical interpretation of the snow cover (SC) – AO correlations. The finding that the decadal correlation between snow cover and AO indices has varied widely with time, between sometimes negative and sometimes positive values for all the five snow cover indices studied (Figure 7), allows for two diverging scenarios:

1. There is a real connection between autumn Eurasian SC and winter AO, but this connection varies with the changing background state of the climate system (e.g., low-frequency variations in SST). The connection may be causal (autumn SC affecting winter AO), or it may reflect a common underlying factor (e.g., interannual variations in SST) that affects both the autumn SC and the winter AO.

2. The connection between autumn Eurasian SC and winter AO is a statistical artefact. When the sample size is small, as it is for 10-year correlations of detrended SC and AO indices, strong correlations sometimes occur by pure chance, occasionally reaching the threshold of statistical significance.

The manuscript does not discuss these two possibilities explicitly, but the spirit of discussion appears to favour the first explanation. However, the results in Figure 7 and Table 3 call for caution. In Table 3, the proportion of decadal correlations that reach the 10 % significance level varies from 4.2 % (SAI_NOV) to 14.1 % (SC_10). A pure chance would give 10 % on the average, but with a substantial variation around this expected number.

I explored the potential importance of chance in creating apparently significant correlations by a Monte Carlo method. I created 179-year time series of pseudo-SC and pseudo-AO indices from normally distributed random numbers with no serial correlation (thus, pure white noise). Then I calculated the 170-year time series of running 10-year correlation between the two indices, after detrending them in the same way as in the manuscript, and further counted the proportion of correlations that exceeded the threshold for 10 % statistical significance. I repeated this 100 000 times.

Within this large Monte Carlo sample, the proportion of statistically significant correlations was 14.1% or larger in 19.9 % of cases. Thus, this fraction of significant correlations could easily be explained by chance. The largest fraction of significant negative correlations (11.8 % for SC_09) turned out to be slightly more unusual, occurring in 5.9 % of the generated time series. However, the probability of getting either at least 11.8 % of negative or at least 11.8 % of positive correlations was nearly twice as large (11.6 %). Finally, the probability of getting a large fraction of significant correlations for at least one of the five SC indices is much larger than that for any specific index alone.

Thus, based on the results presented in this manuscript, one cannot reject the null hypothesis that Eurasian autumn SC has no impact on winter AO at all. This is an important result as such.

In the regional analysis, larger fractions of significant correlations are found, occasionally more than 20 % (Figure 8). The Monte Carlo tests described above indicate a chance of only 1 / 300 of getting either at least 20 % of significantly positive or at least 20 % of significantly negative correlations, provided that full 179 years of data are available (which, however, is not the case for all regions). Again, however, the probability of getting such a large fraction of significant correlations for some of the regions and some of the five SC indices is much larger than that for a single region and a single

SC index. Therefore, similar statistical concerns may apply to them as for the whole-Eurasia SC indices in Figure 7 and Table 3.

Obviously, absence of evidence is not evidence of absence. Some connection between Eurasian autumn SC and winter AO may exist, even if it is not convincingly seen in the correlation analysis. However, the lack of clear statistical evidence suggest that this connection is weak, explaining at best a modest fraction of the AO variability.

My recommendations for improving the statistical analysis in Section 4.2 are as follows:

1. Explicitly acknowledge the fact that much of the correlations and their apparent temporal variation may be due to pure chance.

Based on the statistics from the new 'scrambled data' methodology and new Eurasia region, the revised Table 3 is shown below. This demonstrates that the proportion of decades with a significant SC-AO relationship for Eurasia as a whole is statistically high or low for SCI_09 and SCI_10 for negative and positive relationships with the AO, respectively. In addition, the proportion of decades with a positive SAI_11-AO relationship is significantly small. Therefore, it appears likely that Eurasian snow cover does have some influence on the AO and the relationship is not purely stochastic. This is also the conclusion of Wegmann et al. (2020), who examined SC-AO relationships over the 20[th] Century between November SC and the NAO rather than AO and stated that, "We find evidence for a negative NAO-like signal after November … which is valid throughout the last 150 years."

Wegmann, M., Rohrer, M., Sanolaria-Otín, M., and Lohmann, G.: Eurasian autumn snow link to winter North Atlantic Oscillation is strongest for Arctic warming periods, Earth Syst. Dynam., 11, 509–524, https://doi.org/10.5194/esd-11-509-2020, 2020.

However, I will add the following sentence at an appropriate point, "We note that, despite the data providing statistical evidence for a connection between some autumn SC indices and the following winter AO, the possibility remains that the temporal variations in the strength of the SC-AO relationships are due to pure chance."

**Table 3.** Proportion of decades that the 20CRv3 SC indices have positive and negative correlations with the 20CRv3 AO, and the proportion that are statistically significant based on a one-tailed test ($p < 0.10$). The significance of the proportion of decades with a significant SC-AO correlation is determined based on a probability distribution function derived from 10000 scrambled AO time-series using a one-tailed test for each sign of correlation: *** is $p < 0.01$, ** is $p < 0.05$, * is $p < 0.10$. The period covered is 1831-40 to 2006-2015.

| Snow Cover Index | Positive (%) | Negative (%) | Pos. (significant) (%) | Neg. (significant) (%) |
|---|---|---|---|---|
| SCI_09 | 32.35 | 67.65 | 2.35** | 27.65*** |
| SCI_10 | 48.82 | 51.18 | 4.12* | 22.35** |
| SCI_11 | 52.94 | 47.06 | 11.18 | 10.00 |
| SAI_OCT | 38.82 | 61.18 | 11.18 | 13.53 |
| SAI_NOV | 47.06 | 52.94 | 2.94** | 9.41 |

2. Shorten the discussion of the details in this section.

This will be achieved by reducing the number of subregions analysed in Fig. 8, as suggested by the reviewer, and primarily focusing the discussion on those regions with statistically significant high and low frequencies of significant decadal relationships for opposite signs of the SC-AO relationship in question (see #3 below).

3. Make the significance testing more rigorous. For the results in Figure 7 and Table 3, also report the statistical significance of the proportion of statistically significant correlations. Going beyond the simple Monte Carlo method described above, another choice could be tests in which the connection between SC and AO is broken by randomly scrambling the years of the AO index. For the regional analysis in Fig. 8, only show in colour those regions where the proportion of significant correlations is statistically significant and report the proportion of such regions in the figure itself or in a table.

Many thanks for the excellent suggestion of scrambling the AO index in order to determine the statistical significance of the observed frequency of decadal SC-AO relationships. Following on from some sensitivity tests, I have undertaken this 10000 times for each SC-AO relationship. Importantly, this methodology also allows an examination of whether the frequency of decades with significant correlations is smaller as well as larger than one might expect through chance (one-tailed test). Thus, if the observed frequency is both significantly high for one sign of the SC-AO relationship and significantly low for the other sign then it is likely to be a more robust relationship than one in which the frequency is only significant for one sign. This new methodology has led to revised Figs. 8 (see below) and 9, Table 3 and additional new figures and tables. In combination with the previous analyses in the references suggested by Reviewer 3, using this new methodology has led to a different emphasis in the SC-AO results section and Discussion.

I will add the following text in the methodology and include an example plot in the Supplementary Material:

"To test whether the frequency of decades with a statistically significant SC-AO relationship was itself significant (either high or low), the AO time-series was scrambled 10000 times and probability distribution functions (PDFs) of the frequency of both significant positive and negative SC-AO relationships (based on a one-tailed test at $p < 0.10$) derived (see Fig. SX for an example). As might be expected, the PDFs for positive and negative relationships were very similar but not identical. The observed frequency value was then compared to the appropriate PDF to determine its level of statistical significance if any."

[Figure]

Figure SX. Probability distribution functions of the frequency of statistically significant ($p < 0.10$) decadal SCI_09-AO correlations based on 10000 scrambled AO time-series for (a) positive and (b) negative relationships. The dashed blue lines represent the values of statistical significance ($p < 0.10$, $p < 0.05$, $p < 0.01$) for high and low values. The red line is the value derived from the 20CRV3 data.

Based on the new Fig. 8 (see response to #14 below), the revised Section 4.2 will focus on regions of Eurasia that have both high and low statistically significant frequencies of significant decadal SC-AO correlations of different signs. Principally, these are (i) SCI_09 for north-east Eurasia (120-180°E, 60-70°N) and (ii) SCI_11 for south-west Eurasia (30-60°E, 40-55). The statistics are summarised in the new table and the figure below, the latter demonstrates some interesting decadal variability in the time-series of both SC-AO relationships that can be examined further in comparison to potential external forcing factors.

**Table New.** Proportion of decades that the 20CRv3 SC indices have positive and negative correlations with the 20CRv3 AO, and the proportion that are statistically significant based on a one-tailed test ($p < 0.10$): *** is $p < 0.01$, ** is $p < 0.05$, * is $p < 0.10$. The period covered is 1831-40 to 2006-2015.

| Snow Cover Index | Positive (%) | Negative (%) | Pos. (significant) (%) | Neg. (significant) (%) |
|---|---|---|---|---|
| SCI_09_NE | 26.47 | 73.53 | 1.76** | 27.64*** |
| SCI_11_SW | 76.47 | 23.53 | 38.34*** | 4.12* |

[Figure]

**Figure New.** Running decadal correlations between the SCI_09_NE and SCI_11_SW and winter AO. The dashed horizontal lines of the same colour indicate significance levels at $p < 0.10$ for a one-tailed test.

Apart from the statistical analysis, the manuscript is in reasonable shape. However, as described in the detailed comments below, there is room for improvement in some of the figures and some pieces of the text, and a few details in the methodology. There are also some typos and peculiarities in the wording.

**DETAILED COMMENTS**

1. Replace "positive (negative) relationships" with "statistically significant (positive) negative decadal correlations".

   Done

2. Summing of correlations makes no mathematical sense. For example, if both A and B are perfectly correlated with C (r = 1), the correlation of A + B with C is still 1 and not 2.

   It was not actually supposed to make 'mathematical sense', simply a qualitative device to demonstrate that when the SC-AO relationships of the two regions were of the same sign it gave a 'strong' SC-AO relationship for Eurasia as a whole. Note that following the new methodology this result has been removed anyway as the SAI_OCT dipole is no longer the principal SC-AO relationship.

3. L23-24. A new study suggests an underlying ratio of three when internal climate variability is filtered out from the Arctic temperature time series (Zhou et al.., 2024, Nature Geoscience, doi: 10.1038/s41561-024-01441-1).

   This reference will be added, and the sentence changed to, "It has long been known that the Arctic has been warming significantly faster than the global average, but recent studies have revealed that this 'Arctic amplification' is even greater than previously thought, being a factor of three to four over the past half-century (Rantanen et al., 2022; Zhou et al., 2024).

4. L37-39. A study published in 2002 gives no information of the recent decades, but rather the late 20th century.

   Indeed: the sentence will be amended to, "However, Kitaev et al. (2002) suggested that in the late 20th Century …"

5. at the end of November?

   Yes, this will be amended in the Fig. 1 caption.

6. L83-84. Please name explicitly the emission scenarios that lead to a -10 % vs. -38 % decrease in SC.

The sentence will be changed to, " … they varied from stabilization at ~2060 at −10 ± 5% (relative to 1995-2014) for Shared Socioeconomic Pathway (SSP)123 to continuous and ongoing SC losses that reach −38 ± 10% by 2100 in the least optimistic climate change scenario (SSP585)."

7. L155-156. It would seem better to explicitly exclude the fully oceanic regions from the analysis, as well as the regions with very little land (say less than 20 %).

This has been done in the new analyses. Eurasia is now defined as 35-75°N rather than 35-80°N — all corresponding analyses have been redone and the text changed as appropriate — and an additional six subregions have been removed from the individual analysis, three of which were not included before due to being wholly ocean.

The sentence will be changed to, "In order to examine spatial variability in the SC indices and their relationship to the AO, this region was subdivided into 40 areas, each of 5° latitude by 30° longitude. Note that some of these subregions actually contain little or no SC data as they are predominantly or wholly ocean (Fig. 2) and thus only 34 subregions are analysed individually."

8. Figure 2. The dashed lines are barely visible. Darker blue or thicker lines would be preferable.

This has been done: dark blue for Eurasia and thicker cyan lines to denote the subregions.

[Figure]

**Figure 2.** Red dots are the location of the 133 stations with daily snow depth observations used to validate ERA5. The thick blue line shows Eurasia, as defined in this study, while the dashed cyan lines denote the 34 subregions (5° latitude by 30° longitude) analysed individually.

9. I did not understand this. If SC is either 0 % or 100 % in all days of the month, there is no change with time and thus SAI = 0.

I agree that this could have been phrased better: the sentence now reads, "Trends were only calculated when at least 50% of potential data were available: 'missing' reanalysis data result from a

region either having 0% or 100% SC on all days in a month, giving an SAI of zero and thus making trend analysis problematical if there are a high proportion of such data."

10. Figure 5. The dashed line in (a) are very faint and could be made thicker. Also, there is a slight mismatch in the logic of colours between (a) and (b), since red is used for September in (a) but for October in (b)

All completed.

[Figure]

**Figure 5.** Annual Eurasian SC index data from 20CRv3. (a) SCI_09 (blue), SCI_10 (red) and SCI_11 (brown); (b) SCA_OCT and (c) SCA_NOV: annual data (dotted lines) and running decadal means (full lines) Black lines show trends for 1836-2015, 1921-1965 and 1966-2015.

11. There is very little land in this subregion and thus no reason to pay any attention to it. It might make better sense to totally omit such strongly ocean-dominated regions from the analysis (cf. comment 7 above).

Please see response to comment 7.

12. Very little land in this region as well!

Please see response to comment 7.

13. Figure 7. Please include the index acronyms (SC_09, SC_10, SC_11, SAI_OCT, SAI_NOV) directly within the figure panels, e.g. in their top-left corners.

Completed.

[Figure]

[Figure]

**Figure 7.** Running decadal correlations between snow cover indices and winter AO: (a) SCI_09; (b) SCI_10; (c) SCI_11; (d) SAI_OCT; (e) SAI_NOV. Snow cover indices from 20CRv3 are shown and ERA5 (2003-12 to 2012-21) as red and blue lines, respectively. Correlations with winter AO from CPC and derived from 20CRV3 are shown as full and dashed lines, respectively. The black dashed horizontal lines indicate significance levels at *p* < 0.10 and *p* < 0.05 for each decade considered independently.

14. Figure 8. As discussed in the General comments, it could be better to only use colours for those regions where the fraction of statistically significant decadal correlations is statistically significant at least at the 10 % level.

Completed

[Figure]

**Figure 8.** The frequency of decades with statistically significant correlations with the AO (*p* < 0.10) from the 20CRv3 data (1831-1840 to 2005-2014) for the five SC indices. Frequencies for positive correlations are shown on the left in red and negative correlations on the right in blue. Hatched subregions are where no value was calculated because data availability was less than 50%.

15. Beware of the caveats of multiple testing. The probability that a high fraction of significant correlations is found in some of many areas just by pure chance is far larger than the probability that this happens in any single region.

I will add something along the lines of, "We note that the statistical power of computing the correlations for the subregions separately is potentially compromised by the discrete nature of each test and ignores the fact that the fraction of true null hypotheses is unknown and that some false null hypotheses may be rejected (e.g., Wilks, 2006). Therefore, we focus on those subregions with statistically significant frequencies of SC-AO correlations for both signs of the relationship as likely having the most robust relationships

Wilks, D. S.: On "field significance" and the false discovery rate, J. Appl. Met. Climatol., 45, 1181–1189, https://doi.org/10.1175/JAM2404.1, 2006.

16. What is the correlation of SAI_OCT(west) and SAI_OCT(east)?

Following the revised analysis this comment is no longer relevant.

17. L555-559 and Figure 10. Summing of correlations makes no mathematical sense (cf. Comment 2). In contrast to correlations, covariances are additive, but only if the SAI indices are expressed in absolute (area) units rather than in normalized form. Thus, one option would be to make a plot that shows the absolute covariances of SAI_Oct(East), SAI_Oct(West) and SAI_OCT(East + West) with AO. Another option is to omit Figure 10.

Please see response to comment 2.

L591-592. The interpretation would be this if Figures 11c (11d) showed the correlation between SAI_OCT in west (east) and the 850 hPa geopotential height. However, what is shown is the correlation between the SAI_OCT – AO_winter *correlation* and 850 hPa geopotential height. Modify accordingly.

Following the revised analysis this comment is no longer relevant.

18. L657-658. a trend from negative to positive correlations with winter AO?

Following the revised analysis this comment is no longer relevant.

19. L679-682. What is the correlation between SAI_NOV(west) and AO for the whole period 1836-2015? What is the corresponding correlation between SAI_NOV(east) and AO?

Following the revised analysis this comment is no longer relevant.

**TYPOS ETC.**

1. momentum or moisture? Changed to "… upwelling of longwave radiation and turbulent heat fluxes into the atmosphere …"

2. there have been Corrected, twice!

3. either SAI index? Yes, corrected.

4. Snow cover indices from 20CRv3 and ERA5 ( ) are shown as red and blue lines Yes, corrected.

5. Caption of Table 3. Check the years. The numbers in the table suggest a sample size of exactly 170. Yes, caption corrected.

6. October and November? Yes, corrected.

---

## Author Comment (AC2)

**Reviewer 2**

I thank the reviewer for their comments.

This study uses a long snow cover data to examine the Eurasian autumn snow cover change and its relation with AO. This is an important and interesting topic because of the critical effect of snow cover on the various components of the earth system especially under the warming climate. Overall, this paper has clear structure and the results are well supported by the figures and tables. However, I think the paper lacks impressive findings. The characters of snow cover change and the snow-AO relationship have been investigated by numerous previous studies. Although author uses the long data to explore these issues, there is no new thing in comparison with other studies. Therefore, I would recommend rejecting it for publication in TC.

Considering the comments of all three reviewers, the emphasis for the revised paper will be shifted to how the changes in the autumn SC indices and their relationship to the AO derived from the adjusted 20CRv3 data compares to previous studies. While not necessarily 'impressive', I would argue that the revised paper will achieve the following important results:

1. Ascertain the accuracy of 20CRv3 snow depth data and whether it is an improvement over previous versions. This will help prospective users of the data decide whether it is appropriate for their research needs. The pair of figures below, which are a direct comparison with snow depth from observations, suggest that the high bias in 20CRv2c is indeed carried through to 20CRv3.

[Figure]

**Figure SD.** Bias of the median October-November snow depth in 20CRv2c (top) and 20CRv3 (bottom) from 125 stations

2. Examine the robustness of the relationship between the AO and the two Han & Sun (2018) SC indices — an October index and a November dipole index — in the 20CRv3 data. An initial analysis, based on the scrambled AO methodology proposed by Reviewer 1, suggests that this October SC-AO relationship is not robust at all, while SC in only one of the two regions that comprise the November dipole SC index appears significantly correlated to the AO more frequently than chance (see Table HS below).

**Table HS.** Proportion of decades that the Han & Sun (2018) SC indices derived from 20CRv3 have positive and negative correlations with the 20CRv3 AO, and the proportion that are statistically significant based on a one-tailed test ($p < 0.10$): *** is $p < 0.01$, ** is $p < 0.05$, * is $p < 0.10$. The period covered is 1831-40 to 2006-2015.

| Snow Cover Index | Positive (%) | Negative (%) | Pos. (significant) (%) | Neg. (significant) (%) |
|---|---|---|---|---|
| Oct_SNOWI | 62.36 | 37.64 | 12.94 | 11.76 |
| Nov_SNOWI | 31.18 | 68.82 | 5.88 | 32.94*** |
| Nov_SNOWI (west) | 75.29 | 24.71 | 32.94*** | 4.12* |
| Nov_SNOWI (east) | 34.71 | 65.29 | 7.65 | 18.72 |

3. Examine how the temporal variability of the correlation between these indices and the AO compares to 20CRv2c, for example, as demonstrated by Wegmann et al. (2020) in their Fig. 7b: an equivalent plot using 20CRv3 is shown below. Comparing the two plots indicates a broad similarity, with statistically significant negative SC-AO relationships in the 1920s and the present. Differences include some short periods of positive correlations in the 20CRv3 plot that are not present in 20CRv2c. Moreover, being able to extend the time series back further using 20CRv3 reveals that periods of positive correlation were actually dominant during the mid 19[th] Century, thus indicating that the negative Han & Sun (2018) November dipole-AO relationship is less temporally invariant than previously thought.

[Figure]

**Figure W.** Running 21-year correlations between the Han & Sun (2018) November dipole index and winter AO. The dashed horizontal lines of the same colour indicate significance levels at $p < 0.10$ for a one-tailed test.

4. Introduce an apparently robust SC-AO relationship with the September SC in the north-east of Eurasia (120-180°E, 60-70°N) that has not been described previously. For this region, SCI_09 has both a statistically significantly low frequency of decades with a positive AO correlation and vice versa for a negative AO (see revised Fig. 8 below).

[Figure]

**Figure 8.** The frequency of decades with statistically significant correlations with the AO ($p < 0.10$) from the 20CRv3 data (1831-1840 to 2005-2014) for the five SC indices. Frequencies for positive correlations are shown on the left in red and negative correlations on the right in blue. Hatched subregions are where no value was calculated because data availability was less than 50%.

5. Demonstrate that in 20CRv3 using SC in south-west Eurasia (30-60°E, 40-55) provides a more robust SCI_11-AO relationship than the Han & Sun (2018) Nov_SNOWI index (compare Table_new below with Table HS above); i.e. it has a higher frequency of decades with a significant SC-AO relationship.

**Table_new.** Proportion of decades SC indices derived from 20CRv3 have positive and negative correlations with the 20CRv3 AO, and the proportion that are statistically significant based on a one-tailed test ($p < 0.10$): *** is $p < 0.01$, ** is $p < 0.05$, * is $p < 0.10$. The period covered is 1831-40 to 2006-2015.

| Snow Cover Index | Positive (%) | Negative (%) | Pos. (significant) (%) | Neg. (significant) (%) |
|---|---|---|---|---|
| SCI_09_NE | 26.47 | 73.53 | 1.76** | 27.64*** |
| SCI_11_SW | 76.47 | 23.53 | 38.34*** | 4.12* |

Han, S., and Sun, J.: Impacts of autumnal Eurasian snow cover on predominant modes of boreal winter surface air temperature over Eurasia, J. Geophys. Res. Atmos., 123, 10076–10091, https://doi.org/10.1029/2018JD028443, 2018.

Wegmann, M., Rohrer, M., Sanolaria-Otín, M., and Lohmann, G.: Eurasian autumn snow link to winter North Atlantic Oscillation is strongest for Arctic warming periods, Earth Syst. Dynam., 11, 509–524, https://doi.org/10.5194/esd-11-509-2020, 2020.

Major issues

This is a quite long paper. The author has put a lot of efforts into the Introduction, Data and Method parts, while the Results part seems a little hasty. Also, I cannot easily get the point from the lengthy introduction. I suggest the author to delete the text that unrelated to the main topic of your study and put some methods into the supplementary materials.

As neither of the other reviewers mention this issue, I would prefer to keep the Introduction as it is. However, if the revised paper shifts emphasis as proposed then there is the potential to remove the paragraph about previous modelling studies.

Form figure 7 and table 3, I find that there is hardly any significant correlation between snow and AO. Does it imply that the Eurasian snow cover has little influence on the AO, or the snow-AO relation is a stochastic event?

Based on the statistics from the new 'scrambled data' methodology and new Eurasia region suggested by Reviewer 1, the revised Table 3 is shown below. This demonstrates that the proportion of decades with a significant SC-AO relationship for Eurasia as a whole is statistically high and low for SCI_09 and SCI_10 for negative and positive relationships with the AO, respectively. In addition, the proportion of decades with a positive SAI_11-AO relationship is significantly small. Therefore, it appears likely that Eurasian snow cover does have some influence on the AO and the relationship is not purely stochastic. This is also the conclusion of Wegmann et al. (2020), who examined SC-AO relationships over the 20[th] Century and between November SC and the NAO rather than AO and stated that, "We find evidence for a negative NAO-like signal after November … which is valid throughout the last 150 years."

Wegmann, M., Rohrer, M., Sanolaria-Otín, M., and Lohmann, G.: Eurasian autumn snow link to winter North Atlantic Oscillation is strongest for Arctic warming periods, Earth Syst. Dynam., 11, 509–524, https://doi.org/10.5194/esd-11-509-2020, 2020.

**Table 3.** Proportion of decades that the 20CRv3 SC indices have positive and negative correlations with the 20CRv3 AO, and the proportion that are statistically significant based on a one-tailed test ($p < 0.10$). The significance of the proportion of decades with a significant SC-AO correlation is determined based on a probability distribution function derived from 10000 scrambled AO time-series using a one-tailed test for each sign of correlation: *** is $p < 0.01$, ** is $p < 0.05$, * is $p < 0.10$. The period covered is 1831-40 to 2006-2015.

| Snow Cover Index | Positive (%) | Negative (%) | Pos. (significant) (%) | Neg. (significant) (%) |
|---|---|---|---|---|
| SCI_09 | 32.35 | 67.65 | 2.35** | 27.65*** |
| SCI_10 | 48.82 | 51.18 | 4.12* | 22.35** |
| SCI_11 | 52.94 | 47.06 | 11.18 | 10.00 |
| SAI_OCT | 38.82 | 61.18 | 11.18 | 13.53 |
| SAI_NOV | 47.06 | 52.94 | 2.94** | 9.41 |

How to calculate the running decadal mean and correlation in figure 5 and 7? How long is the sliding window? If you use different sliding window, will it change the existing results?

The sliding window is 10 years as we are interested in decadal means and correlations. This will now be explicitly stated in the appropriate figure captions. Figure W2 below provides an example of changing the length of the sliding window: it represents running correlations between the Han & Sun (2018) November dipole index and winter AO.

[Figure]

**Figure W2.** Running 10-year (blue) and 21-year (red) correlations between the Han & Sun (2018) November dipole index and winter AO. The dashed horizontal lines of the same colour indicate significance levels at $p < 0.10$ for a two-tailed test.

As expected, this figure indicates that the longer periods where the SC-AO relationship is statistically significant are similar for both sliding windows; e.g., the period of negative correlation centred around 1925. Of course, the shorter window has more temporal detail, such as the two short periods of statistically significant positive correlations towards the start of the time series, which are not seen in the longer window. However, overall, the principal outcomes are independent on the window length.

Line 160: Why reverse the sign of SAI index? It is contrary to common sense and makes readers difficult to follow.

This follows from the work of Cohen & Jones (2011) and Peings et al. (2013). It was originally done in Cohen & Jones (2011) so that an SAI time series could be easily compared to a winter AO time series: there was a strong negative correlation between the rate of snow advance and the subsequent winter AO in the period encompassed by the earlier work (cf. their Fig. 1). For consistency, I have kept the same definition as the previous literature but would be happy to change the sign of the SAI indices if the editor thinks it appropriate.

---

## Author Comment (AC3)

**Reviewer 3**

I thank the reviewer for their constructive comments and especially for (i) the suggestion of the key references that I was unaware of and (ii) the suggestion of focusing on the quality of the 20CRv3 data, which I would like to meet 'half-way' in a revised paper (see final comment).

GENERAL REMARKS

In the manuscript "Spatial and temporal changes in autumn Eurasian snow cover and its relationship with the Arctic Oscillation" the author uses long-term reanalysis data to subdivide the correlations between autumn snow cover and winter AO sign regionally. This topic is of great importance and fits the scope of the journal.

That said, I am really torn about the quality of this manuscript. I really want to love it, but I find myself wondering about the goal and the impact.

In general, I agree with the previously published reviews that the results are not substantially new and it is difficult for me to see the key point of the exercise.

However, I can see many positive and useful parts in this paper. I think it would be very helpful for the community to have an updated evaluation of SD in 20CRv3, versus the older products and stations, kind of like an update of the cited Wegmann et al 2017 study with the nice bias removal procedure shown here. And then that can be looked at regionally and you can discuss about why and how 20CRv3 might be improved or not, you could go into more detail for other seasons etc. I think this would be of great value for this community. So basically everything until 4.2 I like.

With 4.2 the science becomes unfocused and diffuse. In the end you have a huge array of mediocre findings in a very complicated form. All of this regionalisation could be, in my opinion, be performed by EOF analysis of the whole field, similar to what Han & Sun did in 2019 (see the link below). You mention and show that there are decadal changes in correlation, but offer no process that might explain that. You remain too superficial for novelty in my eyes. Your KEY result in my eyes, after filtering a lot of text and data for meaning, is the west east anti-correlation pattern between autumn SC and AO. Unfortunately, Han & Sun and Wegmann et al. looked into this already to some degree and you do not put your results in the context of those studies at any point. This is an important oversight.

So what is the goal of this manuscript? Is it to find the most impactful regions for seasonal prediction? Is it to find an explanation for the decadal variability? Is it to evaluate snow in 20CRv3? Is it to do something different, maybe in terms of method, season and region that previous studies did? The goal and thus the evaluation if you reached this goal stays elusive. One major difference to previous

studies is the use of 20CRv3, but again, you did not really show the reader how different that product is from 20CRv2c and 20CRv2.

So my recommendation is that you sharpen your manuscript (agreeing with the two other Reviews) and pinpoint towards your goal. I personally would recommend the goal to be SD evaluation rather than looking at dynamics, simply because arguing for novelty in the dynamics part seems difficult from my side of things.

A few notes on the overall structure:

I think the discussion and conclusion sections are misused.

The discussion section in this manuscript is actually just another result section.

I disagree with this statement as I think the discussion can be a place to expand on the key results. However, following the updated methodology and the new references the discussion will be rewritten with an emphasis on comparing previous work, especially the SC indices of Han & Sun (2018) and the subsequent paper by Wegmann et al. (2020), as updated using the 20CRv3 data over the longer time period (see Table HS below).

**Table HS.** Proportion of decades that the Han & Sun (2018) SC indices derived from 20CRv3 have positive and negative correlations with the 20CRv3 AO, and the proportion that are statistically significant based on a one-tailed test ($p < 0.10$): *** is $p < 0.01$, ** is $p < 0.05$, * is $p < 0.10$. The period covered is 1831-40 to 2006-2015.

| Snow Cover Index | Positive (%) | Negative (%) | Pos. (significant) (%) | Neg. (significant) (%) |
|---|---|---|---|---|
| Oct_SNOWI | 62.36 | 37.64 | 12.94 | 11.76 |
| Nov_SNOWI | 31.18 | 68.82 | 5.88 | 32.94*** |
| Nov_SNOWI (west) | 75.29 | 24.71 | 32.94*** | 4.12* |
| Nov_SNOWI (east) | 34.71 | 65.29 | 7.65 | 18.72 |

Key points from this table:

1. The proportion of decades with a significant Oct_SNOWI-AO relationship is not statistically significant at all.
2. The proportion of decades with a significant negative Nov_SNOWI-AO relationship is statistically significantly high ($p < 0.01$) but not significantly low for a positive relationship.
3. Dividing the November dipole into its two components, it appears that only the western area is statistically related to the AO, which matches the new findings of this work.
4. The SAI_NOV_SW in the new analysis (cf. new Fig. 8 and Table_new below) provides a slightly better predictor for the winter AO than the Han & Sun Nov_SNOWI in the 20CRv3 data in that (i) the proportion of of significant decadal SC-AO correlations is statistically significantly high for one sign of the relationship and significantly low for the other, indicating that the relationship is more temporally stable and (ii) the frequency of time when there is a significant relationship is higher.

[Figure]

**Figure 8.** The frequency of decades with statistically significant correlations with the AO ($p < 0.10$) from the 20CRv3 data (1831-1840 to 2005-2014) for the five SC indices. Frequencies for positive correlations are shown on the left in red and negative correlations on the right in blue. Hatched subregions are where no value was calculated because data availability was less than 50%.

**Table_new.** Proportion of decades SC indices derived from 20CRv3 have positive and negative correlations with the 20CRv3 AO, and the proportion that are statistically significant based on a one-tailed test ($p < 0.10$): *** is $p <$ 0.01, ** is $p < 0.05$, * is $p < 0.10$. The period covered is 1831-40 to 2006-2015.

| Snow Cover Index | Positive (%) | Negative (%) | Pos. (significant) (%) | Neg. (significant) (%) |
|---|---|---|---|---|
| SCI_09_NE | 26.47 | 73.53 | 1.76** | 27.64*** |
| SCI_11_SW | 76.47 | 23.53 | 38.34*** | 4.12* |

And the conclusion is the discussion. However, some key context is missing here.

The conclusion is missing.

Again, I disagree with this statement: while the second half of the conclusion will be changed to reflect the new results, I see no problem with the first sections that briefly outline the methodology used and summarise the principal results regarding temporal changes in the different SC indices.

I was constantly confused by the direction of the correlation. For me more snow to negative AO should be a negative correlation.

As mentioned in the text (line 160) it is only the SAI indices where the sign of the correlation is reversed, and this follows from the work of Cohen & Jones (2011) and Peings et al. (2013). It was originally done in Cohen & Jones (2011) so that an SAI time series could be easily compared to a winter AO time series: there was a strong negative correlation between the rate of snow advance and the subsequent winter AO in the period encompassed by the earlier work (cf. their Fig. 1). The SCI indices are such that more snow leading to a negative AO is indeed a negative correlation. I have kept the same definition as the previous literature but would be happy to change the sign of the SAI indices if the editor thinks it appropriate.

SPECIFIC REMARKS

L12: You point out the novelty of going back to 1836, but here you only mention the last 50 years. I am missing some coherence between the two statements.

Depending on what other changes will be made to the Abstract, I will change the start of the sentence to, "The greatest changes in the snow advance indices (SAI) occur over the past 50 years, in which there has been a slowing …"

L34: "There is often" is too imprecise. When does this dipole happen in the season, when was it strong and why and what is the direction of the gradient? Is it usually a lot of snow in the west and no snow in the east? Does the sign flip interannually? Decadally?

With the addition of the new references, this sentence can be expanded. "There is often an east-west dipole in SC anomalies across Eurasia during November with one pole in eastern Europe and the opposite pole in southeast Siberia and northern Mongolia (e.g., Gastineau et al., 2017; Han and Sun, 2018; Zhang et al., 2023). Typically, there is a marked longitudinal SC gradient with the strength of this varying at multidecadal frequencies, perhaps associated with North Atlantic SST variability (Wegmann et al., 2020).

L45: Upward trend of what? Do you mean positive trend?

I have changed the sentence to, "… the upward trend in CDR-derived SC from 1992 …"

L47: When you say "in reality", what data product do you refer to, if we assume that the CDR is "not reality"?

I have changed the sentence to, "In reality, observations indicate that autumn SC …"

L57: First you mention that the Arctic air is "warmer", then you mention that the cyclones bring "Cold" Arctic air. This is confusing.

I have rewritten the sentence as, "In southern Siberia (50-60°N) these cyclones transport cold air into the region from the north or east and induce negative temperature anomalies that allow SC to persist (Bednorz and Wibig, 2017).

L78: I would summerize Arctic amplification and Climate Change to anthropogenic global warming.

I disagree, as they are not solely a response to anthropogenic activity: e.g., Zhou et al. (2024) state that, "natural variability has substantially modulated the degree of Arctic amplification."

Zhou, W., Leung, L. R., and Lu, J.: Steady threefold Arctic amplification of externally forced warming masked by natural variability, Nat. Geosci., 17, 508–515, https://doi.org/10.1038/s41561-024-01441-1, 2024.

L108: In this context, Wegmann et al 2021 investigated an interesting middle ground, where free running seasonal forecast models were induced with different snow regimes (https://wcd.copernicus.org/articles/2/1245/2021/). They found a small but measurable impact onto the NAO sign.

I will add a new sentence to the revised manuscript, "In the 'middle ground', Wegmann et al. (2021) compared free-running models with different initial SC regimes and demonstrated a small but measurable impact on the winter AO."

L125: Building up on Peings and Douville, Wegmann et al 2020 looked at the link between Eurasian snow and NAO and found that the snow impact is highest during Arctic warm periods, when the Albedo impact of additional snow is strong (https://esd.copernicus.org/articles/11/509/2020/).

I will add an additional sentence at the end of the paragraph, "In contrast, Wegmann et al. (2020) determined that a negative AO signal following a strong November west-to-east SC gradient was valid throughout the last 150 years. Moreover, the strongest correlations occurred during periods of high November snow variability coincident with strong Arctic warming at the beginning and end of the 20th Century." This is something that I aim to analyse further in the new Discussion utilising the longer 20CRv3 data.

L290: I think it is worth mentioning in this manuscript, that the grid size of ERA5 is substantially smaller than than of 20CRv2c, so we would expect better agreement between a station and ERA5 vs a station and 20CRv2c.

The sentence will be rewritten as, "SD in earlier versions of 20CR is known to be biased high (Wegmann et al., 2017) and, in addition, the coarser spatial resolution of 20CR means we might expect poorer agreement than in ERA5."

Also you assume here that the errors of 20CRv2c are carried over to 20CRv3. Until now in the manuscript, you did not show the reader that that is indeed the case. Maybe that would be a worthwhile information for the community to have.

The pair of figures below, which are a direct comparison with snow depth from observations, suggest that the high bias in 20CRv2c is indeed carried through to 20CRv3. I suggest that I make this point and reference these figures in the supplementary material.

[Figure]

**Figure.** Bias of the median October-November snow depth in 20CRv2c (top) and 20CRv3 (bottom) from 125 stations

L694 to end: I think your manuscript would really benefit from discussing the work of Han & Sun 2019 (https://agupubs.onlinelibrary.wiley.com/doi/full/10.1029/2018JD028443), Wegmann et al 2020 and Wegmann et al 2021 regarding this west-east difference in snow impact. In there you will also find some initial explanations for what could be the reason for this behavior. In there you will also find some answers to your last sentence in this manuscript.

I agree completely with this and it's clearly a significant oversight of mine to not include these highly relevant references: I have updated the scope of my journal alerts! Moreover, the previous work of

Han & Sun (2018) and Wegmann et al. (2020) negate some of the novel aspects of the work that I thought I was undertaking. Bearing this in mind, and the comments of Reviewer 3, I propose changing the emphasis of the paper to include an examination of how 20CRv3 compares against the previous version of 20CR (and possibly other datasets) when calculating changes in the Eurasian SC indices and their relationship with the AO. The latter component will include an analysis of the Han & Sun SC indices with the new dataset (see above) and a comparison of their temporal variability with the work of Wegmann et al. (2020). For example, an equivalent plot to their Fig. 7b using 20CRv3 is shown below (Figure W). Comparing the two plots indicates a broad similarity, with statistically significant negative SC-AO relationships in the 1920s and the present. Differences include some short periods of positive correlations in the 20CRv3 plot that are not present in 20CRv2c. Moreover, being able to extend the time series back further using 20CRv3 reveals that periods of positive correlation were actually dominant during the mid 19th Century, thus indicating that the negative Han & Sun (2018) November dipole-AO relationship is less temporally invariant than previously thought.

[Figure]

**Figure W.** Running 21-year correlations between the Han & Sun (2018) November dipole index and winter AO. The dashed horizontal lines of the same colour indicate significance levels at *p* < 0.10 for a one-tailed test.

There is also the 'new' result regarding the statistically significant negative SCI_09-AO relationship in northeast Eurasia (cf. Table_new above), which appears robust, both in terms of having a significantly high frequency of decades with a significant negative SC-AO correlation and a significantly low frequency of decades with a significant positive SC-AO correlation, and which demonstrates some fairly regular decadal variability (see Figure New below) that can be examined further in comparison to potential external forcing factors.

[Figure]

**Figure New.** Running decadal correlations between the SCI_09_NE and SCI_11_SW and winter AO. The dashed horizontal lines of the same colour indicate significance levels at $p < 0.10$ for a one-tailed test.

A new title for the paper might be:

**An examination of changes in autumn Eurasian snow cover indices and the temporal stability of their relationships with the Arctic Oscillation using adjusted 20th Century Reanalysis version 3 snow depth data.**

---

## Author Response (AR1)

**Reviewer 1**

I thank the reviewer for their many constructive comments and especially for the excellent suggestion of using a series of scrambled AO time-series in order to test the statistical significance of the frequency of decades with significant SC-AO correlations.

**GENERAL COMMENTS**

This study first uses the National Oceanographic and Atmospheric Administration (NOAA) 20th Century Reanalysis version 3 (20CRv3), together with some other observational data sets, to investigate trends in the Eurasian autumn (September to November) snow cover in years 1836-2015. Then, the variations in the interannual correlation between Eurasian autumn snow cover and the following winter Arctic Oscillation (AO) index are studied. The study adds to previous knowledge by (i) extending its analysis further back in time than has been done thus far and (ii) by also looking in some detail at the regional characteristics of the trends and the snow cover – AO relationship. In addition to the monthly mean snow cover extent in September, October and November, the rate of advance of snow cover during October and November is also analysed.

The main findings of the study include (i) an overall decreasing trend in snow cover extent in October and November since 1836, but particularly since 1966; (ii) a tendency for anticorrelated variations in the rate of snow cover advance between October and November, which is also reflected in the trends for 1966-2015 (slower advance in October but faster in November, due to a delay in the onset of the snow season); (iii) highly variable decadal correlations between the studied snow cover indices and the winter AO; and (iv) the rates of October snow advance in western and eastern Eurasia have typically opposite correlations with the subsequent winter AO index.

One may question the meaningfulness of analysing snow cover variations back to the year 1836, when little data was available to constrain the 20CRv3 reanalysis. However, I believe that this analysis still makes sense, particularly for the interannual variations. Assimilating surface pressure observations, the reanalysis should at least qualitatively capture the snow cover variability, which is largely forced by atmospheric circulation on the interannual time scale. For the same reason, the reanalysis also likely qualitatively captures the correlation between the autumn snow cover and the winter AO even in the 19th century, even though the number of pressure observations was much smaller than today.

My main concerns about this study relate to Section 4.2. Specifically, the manuscript suffers from a too uncritical interpretation of the snow cover (SC) – AO correlations. The finding that the decadal correlation between snow cover and AO indices has varied widely with time, between sometimes negative and sometimes positive values for all the five snow cover indices studied (Figure 7), allows for two diverging scenarios:

1. There is a real connection between autumn Eurasian SC and winter AO, but this connection varies with the changing background state of the climate system (e.g., low-frequency variations in SST). The connection may be causal (autumn SC affecting winter AO), or it may reflect a common underlying factor (e.g., interannual variations in SST) that affects both the autumn SC and the winter AO.

2. The connection between autumn Eurasian SC and winter AO is a statistical artefact. When the sample size is small, as it is for 10-year correlations of detrended SC and AO indices, strong correlations sometimes occur by pure chance, occasionally reaching the threshold of statistical significance.

The manuscript does not discuss these two possibilities explicitly, but the spirit of discussion appears to favour the first explanation. However, the results in Figure 7 and Table 3 call for caution. In Table 3, the proportion of decadal correlations that reach the 10 % significance level varies from 4.2 % (SAI_NOV) to 14.1 % (SC_10). A pure chance would give 10 % on the average, but with a substantial variation around this expected number.

I explored the potential importance of chance in creating apparently significant correlations by a Monte Carlo method. I created 179-year time series of pseudo-SC and pseudo-AO indices from normally distributed random numbers with no serial correlation (thus, pure white noise). Then I calculated the 170-year time series of running 10-year correlation between the two indices, after detrending them in the same way as in the manuscript, and further counted the proportion of correlations that exceeded the threshold for 10 % statistical significance. I repeated this 100 000 times.

Within this large Monte Carlo sample, the proportion of statistically significant correlations was 14.1% or larger in 19.9 % of cases. Thus, this fraction of significant correlations could easily be explained by chance. The largest fraction of significant negative correlations (11.8 % for SC_09) turned out to be slightly more unusual, occurring in 5.9 % of the generated time series. However, the probability of getting either at least 11.8 % of negative or at least 11.8 % of positive correlations was nearly twice as large (11.6 %). Finally, the probability of getting a large fraction of significant correlations for at least one of the five SC indices is much larger than that for any specific index alone.

Thus, based on the results presented in this manuscript, one cannot reject the null hypothesis that Eurasian autumn SC has no impact on winter AO at all. This is an important result as such.

In the regional analysis, larger fractions of significant correlations are found, occasionally more than 20 % (Figure 8). The Monte Carlo tests described above indicate a chance of only 1 / 300 of getting either at least 20 % of significantly positive or at least 20 % of significantly negative correlations, provided that full 179 years of data are available (which, however, is not the case for all regions). Again, however, the probability of getting such a large fraction of significant correlations for some of the regions and some of the five SC indices is much larger than that for a single region and a single

SC index. Therefore, similar statistical concerns may apply to them as for the whole-Eurasia SC indices in Figure 7 and Table 3.

Obviously, absence of evidence is not evidence of absence. Some connection between Eurasian autumn SC and winter AO may exist, even if it is not convincingly seen in the correlation analysis. However, the lack of clear statistical evidence suggest that this connection is weak, explaining at best a modest fraction of the AO variability.

My recommendations for improving the statistical analysis in Section 4.2 are as follows:

1. Explicitly acknowledge the fact that much of the correlations and their apparent temporal variation may be due to pure chance.

   Based on the statistics from the new 'scrambled data' methodology and new Eurasia region, the revised Table 3 is shown below. This demonstrates that the proportion of decades with a significant SC-AO relationship for Eurasia as a whole is statistically high or low for SCI_09 and SCI_10 for negative and positive relationships with the AO, respectively. In addition, the proportion of decades with a positive SAI_11-AO relationship is significantly small. Therefore, it appears likely that Eurasian snow cover does have some influence on the AO and the relationship is not purely stochastic. This is also the conclusion of Wegmann et al. (2020), who examined SC-AO relationships over the 20[th] Century between November SC and the NAO rather than AO and stated that, "We find evidence for a negative NAO-like signal after November … which is valid throughout the last 150 years."

   Wegmann, M., Rohrer, M., Sanolaria-Otín, M., and Lohmann, G.: Eurasian autumn snow link to winter North Atlantic Oscillation is strongest for Arctic warming periods, Earth Syst. Dynam., 11, 509–524, https://doi.org/10.5194/esd-11-509-2020, 2020.

   However, I have added the following sentence at an appropriate point, "We note that, despite the data providing statistical evidence for a connection between some autumn SC indices and the following winter AO, the possibility remains that the temporal variations in the strength of the SC-AO relationships are due to pure chance."

**Table 3.** Proportion of decades that the 20CRv3 SC indices have positive and negative correlations with the 20CRv3 AO, and the proportion that are statistically significant based on a one-tailed test ($p < 0.10$). The significance of the proportion of decades with a significant SC-AO correlation is determined based on a probability distribution function derived from 10000 scrambled AO time-series using a one-tailed test for each sign of correlation: *** is $p < 0.01$, ** is $p < 0.05$, * is $p < 0.10$. The period covered is 1831-40 to 2006-2015.

| Snow Cover Index | Positive (%) | Negative (%) | Pos. (significant) (%) | Neg. (significant) (%) |
|---|---|---|---|---|
| SCI_09 | 32.35 | 67.65 | 2.35** | 27.65*** |
| SCI_10 | 48.82 | 51.18 | 4.12* | 22.35** |
| SCI_11 | 52.94 | 47.06 | 11.18 | 10.00 |
| SAI_OCT | 61.18 | 38.82 | 13.53 | 11.18 |
| SAI_NOV | 52.94 | 47.06 | 9.41 | 2.94** |

2. Shorten the discussion of the details in this section.

This has been achieved by reducing the number of subregions analysed in Fig. 8, as suggested by the reviewer, and focusing primarily on those regions with statistically significant high and low frequencies of significant decadal relationships for opposite signs of the SC-AO relationship in question (see #3 below).

3. Make the significance testing more rigorous. For the results in Figure 7 and Table 3, also report the statistical significance of the proportion of statistically significant correlations. Going beyond the simple Monte Carlo method described above, another choice could be tests in which the connection between SC and AO is broken by randomly scrambling the years of the AO index. For the regional analysis in Fig. 8, only show in colour those regions where the proportion of significant correlations is statistically significant and report the proportion of such regions in the figure itself or in a table.

Many thanks for the excellent suggestion of scrambling the AO index in order to determine the statistical significance of the observed frequency of decadal SC-AO relationships. Following on from some sensitivity tests, I have undertaken this 10000 times for each SC-AO relationship. Importantly, this methodology also allows an examination of whether the frequency of decades with significant correlations is smaller as well as larger than one might expect through chance (one-tailed test). Thus, if the observed frequency is both significantly high for one sign of the SC-AO relationship and significantly low for the other sign then it is likely to be a more robust relationship than one in which the frequency is only significant for one sign. This new methodology has led to revised Figs. 8 (see below) and 9, Table 3 and additional new figures and tables. In combination with the previous analyses in the references suggested by Reviewer 3, using this new methodology has led to a different emphasis in the SC-AO results section and Discussion.

I have added the following text in the methodology (Sect. 3.2) and include an example plot (Fig. S3) in the Supplementary Material:

"To calculate the significance of the frequency of decades with significant SC-AO correlations, the AO time-series was scrambled 10000 times and probability distribution functions (PDFs) of the frequency of significant positive and negative decadal SC-AO relationships (based on a one-tailed test at $p < 0.10$) were derived separately (see Fig. S3 for an example). The observed frequency was then compared to the appropriate PDF to determine its level of statistical significance."

[Figure]

**Figure S3.** Probability distribution functions of the frequency of statistically significant ($p < 0.10$) decadal SCI_09-AO correlations based on 10000 scrambled AO time-series for (a) positive and (b) negative relationships. The dashed blue lines represent the values of statistical significance ($p < 0.10$, $p < 0.05$, $p < 0.01$) for high and low values. The red line is the value derived from the 20CRv3 data.

Based on the new Fig. 8 (see response to #14 below), the revised Section 4.2 will focus on regions of Eurasia that have both high and low statistically significant frequencies of significant decadal SC-AO correlations of different signs. In particular, we examine the SC-AO relationship for SCI_09 for north-east Eurasia (60-70°N, 120-180°E) (SCI_09_NE) in detail in a new Sect 4.3 as this has not been described previously. The statistics are summarised in Table 4 and Fig. 9 below,

**Table 4.** Proportion of running decades that various SC indices derived from 20CRv3 have positive and negative correlations with the 20CRv3 AO, and the proportion that are statistically significant based on a one-tailed test ($p < 0.10$). SCI_09 NE and SCI_11 SW developed in this study and Nov_SNOWI defined in Han and Sun (2018). The significance of the proportion of decades with a significant SC-AO correlation is determined based on a probability distribution function derived from 10000 scrambled AO time-series using a one-tailed test for each sign of correlation: *** is $p < 0.01$, ** is $p < 0.05$, * is $p < 0.10$. The period covered is 1831-40 to 2006-2015.

| Snow Cover Index | Positive (%) | Negative (%) | Pos. (significant) (%) | Neg. (significant) (%) |
|---|---|---|---|---|
| SCI_09_NE | 26.47 | 73.53 | 1.76** | 27.64*** |
| SCI_11_SW | 76.47 | 23.53 | 38.34*** | 4.12* |
| SAI_OCT_CJ | 54.71 | 45.29 | 12.94 | 12.94 |
| Nov_SNOWI | 31.18 | 68.82 | 5.88 | 32.94*** |
| Nov_SNOWI (west) | 75.29 | 24.71 | 32.94*** | 4.12* |
| Nov_SNOWI (east) | 34.71 | 65.29 | 7.65 | 18.82 |

[Figure]

**Figure 9.** Running correlations between SCI_09 NE with the winter AO derived from 20CRv3. Running decadal correlations (full red line) and 21-year periods (brown dotted line) are shown with similar black lines indicating the appropriate *p* < 0.10 significance level.

Apart from the statistical analysis, the manuscript is in reasonable shape. However, as described in the detailed comments below, there is room for improvement in some of the figures and some pieces of the text, and a few details in the methodology. There are also some typos and peculiarities in the wording.

**DETAILED COMMENTS**

1. Replace "positive (negative) relationships" with "statistically significant (positive) negative decadal correlations".

   No longer relevant as text removed.

2. Summing of correlations makes no mathematical sense. For example, if both A and B are perfectly correlated with C (r = 1), the correlation of A + B with C is still 1 and not 2.

   It was not actually supposed to make 'mathematical sense', simply a qualitative device to demonstrate that when the SC-AO relationships of the two regions were of the same sign it gave a 'strong' SC-AO relationship for Eurasia as a whole. Note that following the new methodology this result has been removed anyway as the SAI_OCT dipole is no longer considered a robust SC-AO relationship.

3. L23-24. A new study suggests an underlying ratio of three when internal climate variability is filtered out from the Arctic temperature time series (Zhou et al.., 2024, Nature Geoscience, doi: 10.1038/s41561-024-01441-1).

   This reference has been added, and the sentence changed to, "While it has long been established that the Arctic has been warming significantly faster than the global average, recent studies have revealed that this 'Arctic amplification' is even greater than previously thought, being a factor of three to four over the past half-century (Rantanen et al., 2022; Zhou et al., 2024).

4. L37-39. A study published in 2002 gives no information of the recent decades, but rather the late 20th century.

   Indeed: the sentence has been amended to, "However, Kitaev et al. (2002) suggested that in the late 20th Century …"

5. at the end of November?

   Yes, this has been amended in the Fig. 1 caption.

6. L83-84. Please name explicitly the emission scenarios that lead to a -10 % vs. -38 % decrease in SC.

   No longer relevant as text removed.

7. L155-156. It would seem better to explicitly exclude the fully oceanic regions from the analysis, as well as the regions with very little land (say less than 20 %).

   I agree. Eurasia is now defined as 35-75°N rather than 35-80°N — all corresponding analyses have been redone and the text changed as appropriate — and an additional six subregions have been removed from the individual analysis, three of which were not included before due to being wholly ocean.

   The text now reads, "In order to examine spatial variability in the SC indices and their relationship to the AO, this region was subdivided into 40 areas (5° latitude by 30° longitude). Note that some of these subregions actually contain little or no SC data as they are predominantly or wholly ocean (Fig. 2) and thus only 34 subregions are analysed individually."

8. Figure 2. The dashed lines are barely visible. Darker blue or thicker lines would be preferable.

   This has been done: dark blue for Eurasia and thicker cyan lines to denote the subregions.

[Figure]

**Figure 2.** Red dots are the location of the 133 stations with daily snow depth observations used to validate ERA5. The thick blue line shows Eurasia, as defined in this study, while the dashed cyan lines denote the 34 subregions (5° latitude by 30° longitude) analysed individually.

9.  I did not understand this. If SC is either 0 % or 100 % in all days of the month, there is no change with time and thus SAI = 0.

    I agree that this could have been phrased better: the sentence now reads, "Trends were only calculated when at least 50% of potential data were available: 'missing' reanalysis data result from a region either having 0% or 100% SC on all days in a month, giving an SAI of zero and thus making trend analysis problematical if there are a high proportion of such data."

10. Figure 5. The dashed line in (a) are very faint and could be made thicker. Also, there is a slight mismatch in the logic of colours between (a) and (b), since red is used for September in (a) but for October in (b)

    All completed.

[Figure]

**Figure 5.** Annual Eurasian SC index data from 20CRv3. (a) SCI_09 (blue), SCI_10 (red) and SCI_11 (brown); (b) SCA_OCT and (c) SCA_NOV: annual data (dotted lines) and running decadal means (full lines) Black lines show trends for 1836-2015, 1921-1965 and 1966-2015.

11. There is very little land in this subregion and thus no reason to pay any attention to it. It might make better sense to totally omit such strongly ocean-dominated regions from the analysis (cf. comment 7 above).

    Please see response to comment 7.

12. Very little land in this region as well!

    Please see response to comment 7.

13. Figure 7. Please include the index acronyms (SC_09, SC_10, SC_11, SAI_OCT, SAI_NOV) directly within the figure panels, e.g. in their top-left corners.

Completed.

[Figure]

**Figure 7.** Running decadal correlations between snow cover indices and winter AO: (a) SCI_09; (b) SCI_10; (c) SCI_11; (d) SAI_OCT; (e) SAI_NOV. Snow cover indices from 20CRv3 are shown and ERA5 (2003-12 to 2012-21) as red and blue lines, respectively. Correlations with winter AO from the CPC and derived from 20CRV3 are shown as full and dashed lines, respectively. The black dashed horizontal lines indicate significance levels at *p* < 0.10 based on a one-tailed test for each sign of correlation for each decade considered independently.

14. Figure 8. As discussed in the General comments, it could be better to only use colours for those regions where the fraction of statistically significant decadal correlations is statistically significant at least at the 10 % level.

Completed.

[Figure]

**Figure 8.** The frequency of decades with statistically significant correlations with the AO (*p* < 0.10) from the 20CRv3 data (1831-1840 to 2005-2014) for the five SC indices. Frequencies for positive SC-AO correlations are shown on the left and negative SC-AO correlations on the right. Only statistically significant frequencies are shaded: high in red and low in blue. Hatched subregions are where no value was calculated because data availability was less than 50%. Subregions with a black dot are those where the frequency is statistically significant using a 21-year smoothing window.

15. Beware of the caveats of multiple testing. The probability that a high fraction of significant correlations is found in some of many areas just by pure chance is far larger than the probability that this happens in any single region.

The following text has been added: "We note that the statistical power of computing the correlations for the subregions separately is potentially compromised by the discrete nature of each test and ignores the fact that the fraction of true null hypotheses is unknown and that some false null hypotheses may be rejected (e.g., Wilks, 2016). Furthermore, in plots showing spatial correlations, the statistical significance is calculated using the false discovery rate (FDR) method described by Wilks (2016): as the correlation data exhibit moderate to strong autocorrelation, $\alpha_{FDR} = 2\,\alpha_{global}$, where $\alpha$ is the significance level."

Wilks, D.: "The stippling shows statistically significant grid points": How research results are routinely overstated. overinterpreted and what to do about it, Bull. Amer. Met. Soc., 97, 2263–2273, https://doi.org/10.1175/BAMS-D-15-00267.1, 2016.

16. What is the correlation of SAI_OCT(west) and SAI_OCT(east)?

Following the revised analysis this comment is no longer relevant.

17. L555-559 and Figure 10. Summing of correlations makes no mathematical sense (cf. Comment 2). In contrast to correlations, covariances are additive, but only if the SAI indices are expressed in absolute (area) units rather than in normalized form. Thus, one option would be to make a plot that shows the absolute covariances of SAI_Oct(East), SAI_Oct(West) and SAI_OCT(East + West) with AO. Another option is to omit Figure 10.

Please see response to comment 2.

L591-592. The interpretation would be this if Figures 11c (11d) showed the correlation between SAI_OCT in west (east) and the 850 hPa geopotential height. However, what is shown is the correlation between the SAI_OCT – AO_winter *correlation* and 850 hPa geopotential height. Modify accordingly.

Following the revised analysis this comment is no longer relevant.

18. L657-658. a trend from negative to positive correlations with winter AO?

Following the revised analysis this comment is no longer relevant.

19. L679-682. What is the correlation between SAI_NOV(west) and AO for the whole period 1836-2015? What is the corresponding correlation between SAI_NOV(east) and AO?

Following the revised analysis this comment is no longer relevant.

**TYPOS ETC.**

1. momentum or moisture? Changed to "… upwelling of longwave radiation and turbulent heat fluxes into the atmosphere …"

2. there have been Corrected, twice!

3. either SAI index? Yes, corrected.

4. Snow cover indices from 20CRv3 and ERA5 ( ) are shown as red and blue lines Yes, corrected.

5. Caption of Table 3. Check the years. The numbers in the table suggest a sample size of exactly 170. Yes, caption corrected.

6. October and November? Yes, corrected.

**Reviewer 2**

I thank the reviewer for their comments.

This study uses a long snow cover data to examine the Eurasian autumn snow cover change and its relation with AO. This is an important and interesting topic because of the critical effect of snow cover on the various components of the earth system especially under the warming climate. Overall, this paper has clear structure and the results are well supported by the figures and tables. However, I think the paper lacks impressive findings. The characters of snow cover change and the snow-AO relationship have been investigated by numerous previous studies. Although author uses the long data to explore these issues, there is no new thing in comparison with other studies. Therefore, I would recommend rejecting it for publication in TC.

Considering the comments of all three reviewers, the emphasis for the revised paper has been shifted from the original. The new methodology suggested by Reviewer 1 — in which the AO time series is scrambled 10000 times to examine whether the frequency of periods with significant SC-AO relationships can be distinguished from having arisen by chance — now forms the basis of this section of the analysis. In addition, the Discussion will now focus on how SC indices derived from the adjusted 20CRv3 data compares to those in previous studies.

While not necessarily 'impressive', I would argue that the revised paper will achieve the following 'new things':

1. Ascertain the accuracy of 20CRv3 snow depth data and whether it is an improvement over previous versions. This will help prospective users of the data decide whether it is appropriate for their research needs. The pair of figures below, which are a direct comparison with snow depth from observations, demonstrate that the high bias in 20CRv2c is indeed carried through to 20CRv3.

[Figure]

[Figure]

**Figure S1.** Bias of the median October-November snow depth in (a) 20CRv2c and (b) 20CRv3. The black horizontal line represents the median bias, the blue boxes show the interquartile range in the bias and the whiskers the larger (smaller) of either the upper (lower) quartile plus (minus) 1.5 times the interquartile range or maximum (minimum) value. The red line is the proportion of the 125 in situ station measurements available for comparison.

2. Introduce an apparently robust SC-AO relationship with the September SC in the north-east of Eurasia (60-70°N, 120-180°E) (SCI_09 NE) that has not been described previously. For this region, SCI_09 has both a statistically significantly low frequency of decades and 21-year periods with a positive AO correlation and vice versa for a negative AO (see revised Fig. 8 below). The SCI_09 NE-AO relationship is explored further in the new Sect. 4.3, which reveals that the September tropospheric height anomalies associated with SCI_09 NE closely match the positive phase of the west Pacific teleconnection pattern.

[Figure]

**Figure 8.** The frequency of running decades with statistically significant correlations with the AO (*p* < 0.10) from the 20CRv3 data (1831-1840 to 2005-2014) for the five SC indices. Frequencies for positive SC-AO correlations are shown on the left and negative SC-AO correlations on the right. Only statistically significant frequencies are shaded: high in red and low in blue. Hatched subregions are where no value was calculated because data availability was less than 50%. Subregions with a black dot are those where the frequency is statistically significant using a 21-year smoothing window.

3. Examine the robustness of the relationship between the AO and the SAI_OCT index of Cohen and Jones (2011) and the November dipole index of Han and Sun (2018) in the longer 20CRv3 data and compare them with two statistically robust relationships described in the current study. An initial analysis, based on the scrambled AO methodology proposed by Reviewer 1, suggests that the October SC-AO relationship is not robust at all, while SC in only one of the two regions that comprise the November dipole SC index appears significantly correlated to the AO more frequently than through chance (see Table 4 below).

**Table 4.** Proportion of running decades that various SC indices derived from 20CRv3 have positive and negative correlations with the 20CRv3 AO, and the proportion that are statistically significant based on a one-tailed test ($p <$ 0.10). SCI_09 NE and SCI_11 SW developed in this study, the SAI_OCR_CJ index of Cohen and Jones (2011) and Nov_SNOWI defined in Han and Sun (2018). The significance of the proportion of decades with a significant SC-AO correlation is determined based on a probability distribution function derived from 10000 scrambled AO time-series using a one-tailed test for each sign of correlation: *** is $p < 0.01$, ** is $p < 0.05$, * is $p < 0.10$. The period covered is 1831-40 to 2006-2015.

| Snow Cover Index | Positive (%) | Negative (%) | Pos. (significant) (%) | Neg. (significant) (%) |
|---|---|---|---|---|
| SCI_09_NE | 26.47 | 73.53 | 1.76** | 27.64*** |
| SCI_11_SW | 76.47 | 23.53 | 38.34*** | 4.12* |
| SAI_OCT_CJ | 54.71 | 45.29 | 12.94 | 12.94 |
| Nov_SNOWI | 31.18 | 68.82 | 5.88 | 32.94*** |
| Nov_SNOWI (west) | 75.29 | 24.71 | 32.94*** | 4.12* |
| Nov_SNOWI (east) | 34.71 | 65.29 | 7.65 | 18.82 |

4. Demonstrate that in 20CRv3 using SC in south-west Eurasia (40-55°N, 30-60°E) provides a slightly more robust SCI_11-AO relationship than the Han & Sun (2018) Nov_SNOWI index (cf. Table 4 above); i.e. it has a higher frequency of decades with a significant SC-AO relationship.

Han, S., and Sun, J.: Impacts of autumnal Eurasian snow cover on predominant modes of boreal winter surface air temperature over Eurasia, J. Geophys. Res. Atmos., 123, 10076–10091, https://doi.org/10.1029/2018JD028443, 2018.

Wegmann, M., Rohrer, M., Sanolaria-Otín, M., and Lohmann, G.: Eurasian autumn snow link to winter North Atlantic Oscillation is strongest for Arctic warming periods, Earth Syst. Dynam., 11, 509–524, https://doi.org/10.5194/esd-11-509-2020, 2020.

Major issues

This is a quite long paper. The author has put a lot of efforts into the Introduction, Data and Method parts, while the Results part seems a little hasty. Also, I cannot easily get the point from the lengthy introduction. I suggest the author to delete the text that unrelated to the main topic of your study and put some methods into the supplementary materials.

I have removed the paragraph talking about future projected changes in Eurasian SC in the CMIP models from the Introduction and significantly reduced the text elsewhere from this section. However, as neither of the other reviewers mention this issue and a significant proportion of the methodology is novel, I have not deleted any significant text from the other sections.

Form figure 7 and table 3, I find that there is hardly any significant correlation between snow and AO. Does it imply that the Eurasian snow cover has little influence on the AO, or the snow-AO relation is a stochastic event?

Based on the statistics from the new 'scrambled data' methodology and new Eurasia region suggested by Reviewer 1, the revised Table 3 is shown below. This demonstrates that the proportion

of decades with a significant SC-AO relationship for Eurasia as a whole is statistically high and low for SCI_09 and SCI_10 for negative and positive relationships with the AO, respectively. In addition, the proportion of decades with a negative SAI_NOV-AO relationship is significantly small. Therefore, it appears likely that Eurasian snow cover does have some influence on the AO at these times and the relationship is not purely stochastic. This is also the conclusion of Wegmann et al. (2020), who examined SC-AO relationships over the 20[th] Century and between November SC and the NAO rather than AO and stated that, "We find evidence for a negative NAO-like signal after November … which is valid throughout the last 150 years."

Wegmann, M., Rohrer, M., Sanolaria-Otín, M., and Lohmann, G.: Eurasian autumn snow link to winter North Atlantic Oscillation is strongest for Arctic warming periods, Earth Syst. Dynam., 11, 509–524, https://doi.org/10.5194/esd-11-509-2020, 2020.

**Table 3.** Proportion of decades that the 20CRv3 SC indices have positive and negative correlations with the 20CRv3 AO, and the proportion that are statistically significant based on a one-tailed test ($p < 0.10$). The significance of the proportion of decades with a significant SC-AO correlation is determined based on a probability distribution function derived from 10000 scrambled AO time-series using a one-tailed test for each sign of correlation: *** is $p < 0.01$, ** is $p < 0.05$, * is $p < 0.10$. The period covered is 1831-40 to 2006-2015.

| Snow Cover Index | Positive (%) | Negative (%) | Pos. (significant) (%) | Neg. (significant) (%) |
|---|---|---|---|---|
| SCI_09 | 32.35 | 67.65 | 2.35** | 27.65*** |
| SCI_10 | 48.82 | 51.18 | 4.12* | 22.35** |
| SCI_11 | 52.94 | 47.06 | 11.18 | 10.00 |
| SAI_OCT | 61.18 | 38.82 | 13.53 | 11.18 |
| SAI_NOV | 52.94 | 47.06 | 9.41 | 2.94** |

How to calculate the running decadal mean and correlation in figure 5 and 7? How long is the sliding window? If you use different sliding window, will it change the existing results?

The sliding window is 10 years as we are primarily interested in decadal means and correlations. This will now be explicitly stated in the appropriate figure captions. However, because there is the potential for running window statistics to give a false impression of low frequency variability because of the inherent autocorrelation in the data we test the robustness of the results by repeating the SC-AO correlation analysis using a 21-year running window as has been used in some previous literature. An SC-AO relationship is only considered 'robust' if the frequencies of periods with a significant SC-AO relationships are significantly high or low using both sliding windows (e.g., see new Fig. 8 above).

Line 160: Why reverse the sign of SAI index? It is contrary to common sense and makes readers difficult to follow.

This follows from the work of Cohen & Jones (2011) and Peings et al. (2013). It was originally done in Cohen & Jones (2011) so that an SAI time series could be easily compared to a winter AO time series: there was a strong negative correlation between the rate of snow advance and the subsequent winter AO in the period encompassed by the earlier work (cf. their Fig. 1). However, as two reviewers have mentioned this issue I have changed the definition of the SAI indices. The following sentence

has been added, "However, unlike Cohen and Jones, the regression coefficient is not multiplied by –1: therefore, in this study a higher SAI corresponds to a faster snow advance and vice versa."

**Reviewer 3**

I thank the reviewer for their constructive comments and especially for (i) providing the key references that I was unaware of and (ii) the suggestion of greater emphasis on the use of the 20CRv3 data. Regarding the latter, the new title for the paper is:

**An examination of changes in autumn Eurasian snow cover and its relationships with the winter Arctic Oscillation using 20[th] Century Reanalysis version 3.**

GENERAL REMARKS

In the manuscript "Spatial and temporal changes in autumn Eurasian snow cover and its relationship with the Arctic Oscillation" the author uses long-term reanalysis data to subdivide the correlations between autumn snow cover and winter AO sign regionally. This topic is of great importance and fits the scope of the journal.

That said, I am really torn about the quality of this manuscript. I really want to love it, but I find myself wondering about the goal and the impact.

In general, I agree with the previously published reviews that the results are not substantially new and it is difficult for me to see the key point of the exercise.

However, I can see many positive and useful parts in this paper. I think it would be very helpful for the community to have an updated evaluation of SD in 20CRv3, versus the older products and stations, kind of like an update of the cited Wegmann et al 2017 study with the nice bias removal procedure shown here. And then that can be looked at regionally and you can discuss about why and how 20CRv3 might be improved or not, you could go into more detail for other seasons etc. I think this would be of great value for this community. So basically everything until 4.2 I like.

With 4.2 the science becomes unfocused and diffuse. In the end you have a huge array of mediocre findings in a very complicated form. All of this regionalisation could be, in my opinion, be performed by EOF analysis of the whole field, similar to what Han & Sun did in 2019 (see the link below). You mention and show that there are decadal changes in correlation, but offer no process that might explain that. You remain too superficial for novelty in my eyes. Your KEY result in my eyes, after filtering a lot of text and data for meaning, is the west east anti-correlation pattern between autumn SC and AO. Unfortunately, Han & Sun and Wegmann et al. looked into this already to some degree and you do not put your results in the context of those studies at any point. This is an important oversight.

So what is the goal of this manuscript? Is it to find the most impactful regions for seasonal prediction? Is it to find an explanation for the decadal variability? Is it to evaluate snow in 20CRv3? Is it to do something different, maybe in terms of method, season and region that previous studies did? The

goal and thus the evaluation if you reached this goal stays elusive. One major difference to previous studies is the use of 20CRv3, but again, you did not really show the reader how different that product is from 20CRv2c and 20CRv2.

So my recommendation is that you sharpen your manuscript (agreeing with the two other Reviews) and pinpoint towards your goal. I personally would recommend the goal to be SD evaluation rather than looking at dynamics, simply because arguing for novelty in the dynamics part seems difficult from my side of things.

A few notes on the overall structure:

I think the discussion and conclusion sections are misused.

The discussion section in this manuscript is actually just another result section.

I think the discussion can be a place to expand on key results. However, following the updated methodology and the new references the discussion has been completely rewritten with an emphasis on comparing SC indices derived from 20CRv3 with those in the literature, especially the SC indices of Cohen and Jones (2011) and Han & Sun (2018) and the subsequent paper by Wegmann et al. (2020). Table 4 (below) summarises the findings.

**Table 4.** Proportion of running decades that various SC indices derived from 20CRv3 have positive and negative correlations with the 20CRv3 AO, and the proportion that are statistically significant based on a one-tailed test ($p < 0.10$). SCI_09 NE and SCI_11 SW developed in this study, the SAI_OCR_CJ index of Cohen and Jones (2011) and Nov_SNOWI defined in Han and Sun (2018). The significance of the proportion of decades with a significant SC-AO correlation is determined based on a probability distribution function derived from 10000 scrambled AO time-series using a one-tailed test for each sign of correlation: *** is $p < 0.01$, ** is $p < 0.05$, * is $p < 0.10$. The period covered is 1831-40 to 2006-2015.

| Snow Cover Index | Positive (%) | Negative (%) | Pos. (significant) (%) | Neg. (significant) (%) |
|---|---|---|---|---|
| SCI_09_NE | 26.47 | 73.53 | 1.76** | 27.64*** |
| SCI_11_SW | 76.47 | 23.53 | 38.34*** | 4.12* |
| SAI_OCT_CJ | 54.71 | 45.29 | 12.94 | 12.94 |
| Nov_SNOWI | 31.18 | 68.82 | 5.88 | 32.94*** |
| Nov_SNOWI (west) | 75.29 | 24.71 | 32.94*** | 4.12* |
| Nov_SNOWI (east) | 34.71 | 65.29 | 7.65 | 18.82 |

Key points from this table:

1. The proportion of decades with a significant SAI_OCT_CJ-AO relationship is not statistically significant at all.
2. The proportion of decades with a significant negative Nov_SNOWI-AO relationship is statistically significantly high ($p < 0.01$) but not significantly low for a positive relationship.
3. Dividing the November dipole into its two components, it appears that only the western area is statistically related to the AO, which matches the findings of Wegmann et al. (2021).

4. In the 20CRv3 data, the SAI_NOV_SW (40-55°N, 30-60°E) index provides a slightly better predictor for the winter AO than the Han & Sun Nov_SNOWI in that (i) the proportion of of significant decadal SC-AO correlations is statistically significantly high for one sign of the relationship and significantly low for the other, indicating that the relationship is more temporally stable and (ii) the frequency of time when there is a significant relationship is higher (cf. Table 4 above).

5.

And the conclusion is the discussion. However, some key context is missing here.

The conclusion is missing.

I have to disagree with this statement: while the second half of the conclusion has been changed to reflect the new results, I see no problem with the first sections that briefly outline the methodology used and summarise the principal results regarding temporal changes in the different SC indices.

I was constantly confused by the direction of the correlation. For me more snow to negative AO should be a negative correlation.

As mentioned in the text it is only the SAI indices where the sign of the correlation is reversed, and this follows from the work of Cohen & Jones (2011) and Peings et al. (2013). It was originally done in Cohen & Jones (2011) so that an SAI time series could be easily compared to a winter AO time series: there was a strong negative correlation between the rate of snow advance and the subsequent winter AO in the period encompassed by the earlier work (cf. their Fig. 1). However, as two reviewers have mentioned this issue I have changed the definition of the SAI indices. The following sentence has been added, "However, unlike Cohen and Jones, the regression coefficient is not multiplied by –1: therefore, in this study a higher SAI corresponds to a faster snow advance and vice versa."

SPECIFIC REMARKS

L12: You point out the novelty of going back to 1836, but here you only mention the last 50 years. I am missing some coherence between the two statements.

I agree. The Abstract now reads, "Across the full span of 20CRv3 there is a small increase in mean September Eurasian SC. In contrast, there have been significant decreases in both October and November SC. Trends over the past 50 years demonstrate a slowing and accelerating of snow advance in October and November, respectively, corresponding to a postponement of SC onset."

L34: "There is often" is too imprecise. When does this dipole happen in the season, when was it strong and why and what is the direction of the gradient? Is it usually a lot of snow in the west and no snow in the east? Does the sign flip interannually? Decadally?

With the addition of the new references, this sentence has been expanded into two, which are now located in two different paragraphs within the Introduction. These are:

"In November there is often a west-east dipole in SC anomalies across Eurasia with one pole in eastern Europe and the other in southeast Siberia and northern Mongolia (e.g., Gastineau et al., 2017; Han and Sun, 2018; Zhang et al., 2023)."

"However, in contrast, Wegmann et al. (2020) determined that a negative AO signal following a strong November west-to-east SC gradient was valid throughout the last 150 years. Moreover, the strongest correlations occurred during periods of high November snow variability coincident with strong Arctic warming at the beginning and end of the 20$^{th}$ Century."

L45: Upward trend of what? Do you mean positive trend?

I have changed the sentence to, "… the upward trend in CDR-derived SC from 1992 …"

L47: When you say "in reality", what data product do you refer to, if we assume that the CDR is "not reality"?

I have changed the sentence to, "In reality, observations indicate that autumn SC …"

L57: First you mention that the Arctic air is "warmer", then you mention that the cyclones bring "Cold" Arctic air. This is confusing.

I have rewritten the sentence as, "In southern Siberia (50-60°N) these cyclones transport cold air into the region from the north or east and induce negative temperature anomalies that allow SC to persist (Bednorz and Wibig, 2017).

L78: I would summerize Arctic amplification and Climate Change to anthropogenic global warming.

I disagree, as they are not solely a response to anthropogenic activity: e.g., Zhou et al. (2024) state that, "natural variability has substantially modulated the degree of Arctic amplification."

Zhou, W., Leung, L. R., and Lu, J.: Steady threefold Arctic amplification of externally forced warming masked by natural variability, Nat. Geosci., 17, 508–515, https://doi.org/10.1038/s41561-024-01441-1, 2024.

L108: In this context, Wegmann et al 2021 investigated an interesting middle ground, where free running seasonal forecast models were induced with different snow regimes (https://wcd.copernicus.org/articles/2/1245/2021/). They found a small but measurable impact onto the NAO sign.

I have added a new sentence to the revised manuscript, "In the 'middle ground', Wegmann et al. (2021) compared free-running models with different initial SC regimes and demonstrated a small but measurable impact on the winter AO."

L125: Building up on Peings and Douville, Wegmann et al 2020 looked at the link between Eurasian snow and NAO and found that the snow impact is highest during Arctic warm periods, when the Albedo impact of additional snow is strong (https://esd.copernicus.org/articles/11/509/2020/).

I have added an additional sentence at the end of the paragraph, "However, in contrast, Wegmann et al. (2020) determined that a negative AO signal following a strong November west-to-east SC gradient was valid throughout the last 150 years. Moreover, the strongest correlations occurred during periods of high November snow variability coincident with strong Arctic warming at the beginning and end of the 20th Century."

L290: I think it is worth mentioning in this manuscript, that the grid size of ERA5 is substantially smaller than that of 20CRv2c, so we would expect better agreement between a station and ERA5 vs a station and 20CRv2c.

An additional sentence is now included: "We note that the coarser spatial resolution of 20CRv3 versus ERA5 is likely also a contributing factor to the offset."

Also you assume here that the errors of 20CRv2c are carried over to 20CRv3. Until now in the manuscript, you did not show the reader that that is indeed the case. Maybe that would be a worthwhile information for the community to have.

The pair of figures below, which are a direct comparison with snow depth from observations, indicate that the high bias in 20CRv2c is indeed carried through to 20CRv3. These have been added to the Supplementary Material (Fig. S1). In addition, the following paragraph has been added to Sect. 3.1:

"SD in earlier versions of 20CR is known to be biased high (Wegmann et al., 2017). To determine whether 20CRv3 is more accurate, both 20CRv2c and 20CRv3 SD data were compared directly to SD measurements at 125 of the meteorological stations employed in this analysis. The eight missing stations are a consequence of the coarser spatial resolution of 20CRv2c, as they are located on the 'wrong side' of the land-sea mask. Time-series of the mean daily October-November (ON) bias for the two versions of 20CR are shown in Fig. S1, which indicates there is no improvement in the high bias in autumn SC in 20CR. In fact, 20CRv3 is slightly worse: the mean SD bias across the whole ON dataset is 6.2 cm and 8.3 cm for 20CRv2c and 20CRv3, respectively."

[Figure]

[Figure]

**Figure S1.** Bias of the median October-November snow depth in (a) 20CRv2c and (b) 20CRv3. The black horizontal line represents the median bias, the blue boxes show the interquartile range in the bias and the whiskers the larger (smaller) of either the upper (lower) quartile plus (minus) 1.5 times the interquartile range or maximum (minimum) value. The red line is the proportion of the 125 in situ station measurements available for comparison.

L694 to end: I think your manuscript would really benefit from discussing the work of Han & Sun 2019 (https://agupubs.onlinelibrary.wiley.com/doi/full/10.1029/2018JD028443), Wegmann et al 2020 and Wegmann et al 2021 regarding this west-east difference in snow impact. In there you will also find some initial explanations for what could be the reason for this behavior. In there you will also find some answers to your last sentence in this manuscript.

I completely agree and it's clearly a significant oversight of mine to not have originally included these highly relevant references: I have updated the scope of my journal alerts! Moreover, the previous work of Han & Sun (2018) and Wegmann et al. (2020) means that some of the analysis undertaken in this work is no longer novel. Bearing this in mind and the comments of Reviewer 2, as mentioned previously the Discussion has been rewritten to include an analysis of the SC indices of Cohen and Jones (2011) and Han & Sun (2018) and the subsequent paper by Wegmann et al. (2020).

For example, an equivalent plot to their Fig. 7b (correlation of the Nov_SNOWI index vs (N)AO) using 20CRv3 is shown in Fig. 12b below. The associated text is, "In detail, the 20CRv3 data lie somewhere between those derived from 20CRv2c and ERA20C: (i) there are periods of significantly negative correlations in both the 1920s and 1990s+ while ERA20C only has the latter and (ii) there is a period of positive correlations at ~1975 while ERA20C has an additional earlier period and 20CRv2c has neither. In the 19[th] Century — earlier than Wegmann et al. (2020) examined — there is an additional period of significant negative correlations at ~1875 and also a period of significant positive SC-AO relationship at the beginning of the 20CRv3 dataset at ~1850. As expected, there is more variability in the periods of significant correlations in the decadally-smoothed data but there are no decades with a significant positive Nov_SNOWI index after 1900, confirming the robustness of this SC-AO relationship after this time."

[Figure]

**Figure 12.** Running correlations between (a) the SAI_OCR_CJ index of Cohen and Jones (2011) and (b) the November dipole index of Han and Sun (2018) with the winter AO all derived from 20CRv3. Running decadal correlations (full red line) and 21-year periods (brown dotted line) are shown with similar black lines indicating the appropriate *p* < 0.10 significance level.

---

## Author Response (AR2)

**Reviewer 1**

I thank the reviewer for their new comments on the revised manuscript.

**GENERAL COMMENTS**

There are several improvements in this revised manuscript compared with the original version. First, the statistical testing of the findings is now more rigorous and the found correlations are more cautiously interpreted. Second, the revised manuscript highlights a possible (although weak) connection between September snow cover in Northeastern Siberia and Winter NAO, which has not been proposed before. Third, a more detailed comparison with earlier studies has been added to Section 5. Finally, some earlier text and figures have been deleted.

I have no remaining major concerns on this study. However, its main value as I see it is still more in the negative than in the positive findings. Rather than demonstrating a clear connection between Eurasian autumn snow cover and winter Arctic Oscillation (AO), the study effectively documents the weakness and ephemeral nature of any such connections.

From the reader point of view, the first half of the manuscript is easy to follow. However, Section 4.2 and to some extent 5 are heavy due to the large number of details reported in the text. Omitting some of the details would therefore be beneficial, even though it is difficult to give point-to-point suggestions of what to omit. Yet, for example, the timing of the individual maxima and minima in the correlation time series generally would not need to be reported, since this is subordinate to the main finding that the correlations have been widely variable.

I have removed several sentences across Sections 4.2, 4.3 and 5.

Although I would like to avoid suggestions that risk the further lengthening of the manuscript, I still make one. To complement the "frequency of significant decadal correlations" analysis in Sections 4.2-5 (which is needed but somewhat complicated), it would be useful to also report the average correlations between the various snow cover indexes and the winter AO, for the whole 1836-2015 period. These could be easily included as an additional column of maps in Figure 8 and as numerical values written directly in the time series plots (Figs. 7, 9 and 12). I leave it to the author to decide whether averages of the running 10-year correlations or the correlations of annual values in the whole 1836-2015 time series are more suitable for this.

This has been completed as requested, with a focus on the average of the running 10-year correlations. In summary, Fig. 8 has been expanded as requested and numerical correlation coefficient values added to Figs. 7, 9 and 12 with additional associated text in Sections 3.2, 4.2, 4.3 and 5.

Before the detailed comments, two generic technical issues:

1. The dashed lines in many of the time series plots (as well as in the map in Fig. 2) appeared very faint in this revised manuscript. This might partly be because the figures have been compressed to too low resolution – an interpretation supported by the fact that the merged pdf file for the revised manuscript was a factor of five smaller than the pdf of the original version. Still, it would be better to make all the dashed lines (including the vertical dashed lines in some of the time series plots!) systematically thicker to avoid this problem.

While I think the majority of the issues with the figures are associated with the compression, I have made the dashed lines thicker (also see comments from the Editor)

2. In several places in Section 4.2, the first year of the analysis is reported as 1831 (or the first decade as 1831-1840) although 20CRv3 is only available from 1836. Please check this throughout.

Thank you for spotting these issues: all corrected.

DETAILED COMMENTS

1. L113. high and low

Done

2. L130. and describe the different

Done

3. L237-238. "multi-sensor" and "system" repeated

Done

4. L371. SCI_09 in brown and SCI_11 in blue?

Done

5. L428-429. in the low frequency of periods positive SCI_10 - AO correlations?

Corrected to, "for the low frequency of periods with positive SCI_10 -AO correlations"

6. L443. from 20CRv3 and ERA5 (2003-12 to 2012-21) are shown as red and blue

Done

7. L478. frequency of significant decadal relationships

Done

8. L540. "situated" is repeated

Done

9. L553. Is this really a "response" to SCI_09 NE anomalies or a signature of the circulation type that causes these anomalies?

'Response' changed to 'contemporaneous change'

10. L560. at high latitudes

Done

11. Figure 10. Add the headings "Surface air temperature", "Precipitation" and "Z(500 hPa)" above the three panels.

Done

12. L704-705. Combine the two incomplete sentences: "… (SCI_09NE), and …".

Done

13. L706. Providing an average correlation of -0.18, SCI_09 NE still appears to be a very weak predictor.

As I have switched to focusing on the mean decadal correlation, the text around this value has been changed as has the value itself (–0.23), which is still significant at $p < 0.01$ based on the Monte Carlo analysis. However, I do agree with the reviewer and have added a couple of new sentences in Sections 4.2 and 5 that are not specific to the SCI_09 NE region but note the ephemeral nature of the strong SC-AO relationships, which means that overall, they are relatively weak predictors of the winter AO. These are:

"However, the mean decadal correlation is only statistically significant for SCI_09: while the value of –0.19 is significant at $p < 0.01$, based on the Monte Carlo analysis, for the complete time series it appears a weak predictor."

"Overall, SCI_09_NE, SCI_11_SW and the Nov_SNOWI dipole index of Han and Sun (2018) can be considered similarly robust across the 180 years of 20CRv3: indeed, the mean decadal correlation coefficients for these three SC indices are –0.23, 0.25 and –0.22, respectively. Thus, while these values all indicate a significant relationship with the following winter AO — at $p < 0.01$, based on the Monte Carlo analysis — their relatively small magnitude demonstrates the ephemeral nature of the relationships"

14. L718. "a simultaneous signature" would be more suitable than "an immediate

response", given that these circulation anomalies are more likely a cause than a consequence of the SC anomalies.

'an immediate response changed to 'a contemporaneous signature'

15. L723-725. I agree with this need of model experiments, even though these experiments would most likely confirm that the response is very weak.

Agreed, but it should provide direct information on the mechanism involved

16. Figure S5 is blurred. Please provide a higher-resolution version.

Yes, not sure what happened there but will double-check the uploaded file this time

**Editor**

- Figure 3: "black horizontal line" (in particular after 1992) and "vertical dashed red line"
- Figure 4: "vertical dashed red lines"
- Figure 5: "dotted lines"
- Figure 7: "dashed lines" and "black dashed horizontal lines"
- Figure 9: "brown dotted line" and "similar black lines"
- Figure 10: purple, green, and yellow contours
- Figure 12: "brown dotted line" and "similar black lines"
- Figure S4: "dashed lines" and "black dashed horizontal lines"
- Figure S5: All the numbers and texts

All changes made as requested